# Therapeutic effects of telomerase in mice with pulmonary fibrosis induced by damage to the lungs and short telomeres

Juan Manuel Povedano[1†], Paula Martinez[1†], Rosa Serrano[1], Águeda Tejera[1], Gonzalo Gómez-López[2], Maria Bobadilla[3,4], Juana Maria Flores[5], Fátima Bosch[6], Maria A Blasco[1]*

[1]Telomeres and Telomerase Group, Molecular Oncology Program, Spanish National Cancer Centre, Madrid, Spain; [2]Bioinformatics Core Unit, Structural Biology and Biocomputing Program, Spanish National Cancer Centre, Madrid, Spain; [3]Roche Pharma Research and Early Development (pRED), Neuroscience, Ophthalmology and Rare Disease, Roche Innovation Center Basel, F. Hoffmann-La Roche Ltd, Basel, Switzerland; [4]Roche Partnering, EIN, F. Hoffmann-La Roche Ltd, Basel, Switzerland; [5]Animal Surgery and Medicine Department, Faculty of Veterinary Science, Complutense University of Madrid, Madrid, Spain; [6]Centre of Animal Biotechnology and Gene Therapy, Department of Biochemistry and Molecular Biology, School of Veterinary Medicine, Autonomous University of Barcelona, Bellaterra, Spain

**Abstract** Pulmonary fibrosis is a fatal lung disease characterized by fibrotic foci and inflammatory infiltrates. Short telomeres can impair tissue regeneration and are found both in hereditary and sporadic cases. We show here that telomerase expression using AAV9 vectors shows therapeutic effects in a mouse model of pulmonary fibrosis owing to a low-dose bleomycin insult and short telomeres. AAV9 preferentially targets regenerative alveolar type II cells (ATII). AAV9-*Tert*-treated mice show improved lung function and lower inflammation and fibrosis at 1–3 weeks after viral treatment, and improvement or disappearance of the fibrosis at 8 weeks after treatment. AAV9-*Tert* treatment leads to longer telomeres and increased proliferation of ATII cells, as well as lower DNA damage, apoptosis, and senescence. Transcriptome analysis of ATII cells confirms downregulation of fibrosis and inflammation pathways. We provide a proof-of-principle that telomerase activation may represent an effective treatment for pulmonary fibrosis provoked or associated with short telomeres.

DOI: https://doi.org/10.7554/eLife.31299.001

*For correspondence:
mblasco@cnio.es

[†]These authors contributed equally to this work

## Introduction

Mammalian telomeres are protective structures at ends of chromosomes that consist of TTAGGG repeats bound by a six-protein complex known as shelterin (*Blackburn, 2001*; *de Lange, 2005*). A minimum length of telomeric repeats is necessary for shelterin binding and telomere protection (*Blackburn, 2001*; *de Lange, 2005*). Telomerase is an enzyme composed of two subunits, the telomerase reverse transcriptase (TERT) and the RNA component (*Terc*), which is used as template for the de novo addition of telomeric repeats to chromosome ends (*Greider and Blackburn, 1985*). Adult tissues, including the stem cell compartments, do not have sufficient telomerase activity to compensate for the progressive telomere shortening associated with cell division throughout lifespan (*Canela et al., 2007*; *Flores et al., 2008*; *Harley et al., 1990*; *Vera et al., 2012*). When telomeres reach a critically short length, this triggers activation of a persistent DNA damage response at

**eLife digest** Idiopathic pulmonary fibrosis (or IPF for short) is a rare disease that scars the lungs. The condition gets worse over time, making it harder and harder to breathe, and eventually leading to death. Patients typically only survive for a few years after being diagnosed with IPF. This is because, as yet, there is no cure; the available treatments only act to lessen the symptoms.

Several risk factors have linked to the development of IPF, among them, the presence of short telomeres. Like the plastic tips on shoelaces, telomeres are protective structures at the ends of chromosomes. Telomeres shorten with age, and when they become too short the cell stops dividing and often dies in a process known as apoptosis. IPF can develop when the telomeres in the cells that repair everyday wear and tear in the lungs (known as ATII cells) become too short. This means that the damage goes unrepaired, triggering an immune reaction and uncontrolled scarring.

Telomerase is an enzyme that can lengthen short telomeres, and Povedano, Martíńez et al. set out to develop a new treatment approach that would use this enzyme to correct the short telomeres, and cure the scarring seen in IPF. Gene therapy was used to introduce the gene for telomerase into mice that had scarring in their lungs due to short telomeres.

Povedano, Martíńez et al. found that, when injected into the mice, the telomerase gene therapy was able to reach ATII cells and could help to heal the lungs. At the level of individual cells, mice treated with telomerase had longer telomeres, meaning that more of their ATII cells stayed alive and kept dividing to regenerate the lung tissue. Consistent with previous studies, the telomerase gene therapy caused no negative side effects in the mice; for example, there was no increased risk of cancer.

These findings may possibly lead to new treatments for those patients suffering from IPF associated with short telomeres. Developing this approach into a clinical trial could in the future benefit many IPF patients who currently have very limited treatment options.

DOI: https://doi.org/10.7554/eLife.31299.002

---

telomeres and the subsequent induction of cellular senescence or apoptosis. Indeed, this progressive shortening of telomeres with increasing age is considered one of the hallmarks of aging both in mice and humans (*López-Otín et al., 2013*). In particular, critical telomere shortening at the stem cell compartments results in the loss of the regenerative capacity of these compartments eventually compromising tissue renewal and homeostasis (*Blasco, 2007*; *Flores et al., 2005*; *Povedano et al., 2015*). Interestingly, the rate of telomere shortening throughout lifespan has been shown to be influenced both by genetic factors (ie., mutations in genes necessary for telomere maintenance) and environmental factors (ie., cigarette smoke has a negative effect) (*Armanios, 2013*; *King et al., 2011*).

In support of critical telomere shortening being a determinant of aging and longevity, increased TERT expression in the context of cancer resistant transgenic mice was sufficient to delay aging and extend mouse longevity by 40% (*Tomás-Loba et al., 2008*). More recently, these findings have been translated into a potential therapeutic strategy by using adeno-associated vectors (AAV) to transiently activate telomerase in adult tissues (*Bär et al., 2014*; *Bernardes de Jesus et al., 2012*). In particular, treatment with *Tert* gene therapy using non-integrative AAV9 vectors of adult mice was able to delay aging and increase longevity by decreasing age-related pathologies such as osteoporosis, glucose intolerance, as well as neuromuscular and cognitive decline. Furthermore, the onset of cancer was also delayed in the *Tert* treated mice (*Bernardes de Jesus et al., 2012*). More recently, AAV9-*Tert* delivery specifically to the heart was sufficient to significantly increase mouse survival and heart function upon myocardial infarction, which was concomitant with decreased fibrosis and increased cardiac myocyte proliferation (*Bär et al., 2014*). These findings support the notion that telomere shortening is at the origin of age-related diseases and that, by delaying or reverting this process with telomerase, it is possible to delay and treat more effectively age-associated diseases, such as heart infarct.

Extreme telomere shortening can occur prematurely in individuals with mutations in telomerase and other telomere maintenance genes causing the so-called telomere syndromes, which include dyskeratosis congenita, aplastic anemia and pulmonary fibrosis, among others (for a review see

[*Armanios and Blackburn, 2012*]). These syndromes are characterized by premature loss of the regenerative capacity of tissues, affecting both high and low proliferation tissues (*Armanios and Blackburn, 2012*; *Holohan et al., 2014*). Among the telomere syndromes, idiopathic pulmonary fibrosis (IPF) is the most common condition associated with telomere dysfunction in humans (*Armanios, 2013*; *Armanios and Blackburn, 2012*). Both familial and sporadic cases have been linked to telomerase mutations, either in *TERT* or *TERC* (*Alder et al., 2008*; *Armanios et al., 2007*). In particular, mutations in *TERT* and *TERC* account for 8–15% of familial and 1–3% of sporadic cases (*Alder et al., 2008*; *Armanios, 2013*; *Armanios et al., 2007*). Interestingly, sporadic cases of IPF, not associated with telomerase mutations, also show shorter telomeres compared to age-matched controls, with 10% of the patients showing telomeres as short as the telomerase mutation carriers (*Alder et al., 2008*). Telomerase mutations have also been found in up to 1% of smokers showing chronic obstructive pulmonary disease (COPD), also leading to abnormally short telomeres (*Stanley et al., 2015*).

Unfortunately, in spite of its prevalence, idiopathic pulmonary fibrosis is still a life-threatening lung degenerative disease, with few available therapeutic options (*King et al., 2011*). As an example, the recently FDA-approved drugs, nintedanib and pirfenidone, show anti-inflammatory and anti-fibrotic activity (*Ahluwalia et al., 2014*; *Karimi-Shah and Chowdhury, 2015*; *King et al., 2014*), and slow IPF progression but are not curative (*Hunninghake, 2014*; *Karimi-Shah and Chowdhury, 2015*; *King et al., 2014*). Indeed, to date, lung transplantation is the only curative therapeutic option in less than 5% of IPF patients with severe disease (*Lama, 2009*). Thus, development of new, more effective, therapeutic strategies aimed against treating the origin of the disease is urgently needed.

An important limitation to the development of new therapeutic strategies has been the lack of appropriate pre-clinical mouse models. Induction of acute pulmonary fibrosis with high doses of bleomycin in mice has been the most widely used preclinical model, although the disease spontaneously reverses in this model after 2–3 weeks (*Mouratis and Aidinis, 2011*). Furthermore, telomerase-deficient mice with short telomeres do not spontaneously develop pulmonary fibrosis (*Alder et al., 2011*), suggesting that additional insults contribute to the disease in addition to the genetic defects. In support of this notion, we recently demonstrated that treatment with low doses of bleomycin (0.5 mg/kg BW), which normally do not lead to pulmonary fibrosis in wild-type mice, however, results in full-blown progressive pulmonary fibrosis in telomerase deficient mice (*Povedano et al., 2015*). Thus, this model shows that short telomeres are at the molecular origin of pulmonary fibrosis and could represent a useful pre-clinical tool to test the challenging hypothesis of whether therapeutic strategies based on telomerase activation maybe effective in the treatment of the disease.

Here, we tested this hypothesis by using a *Tert* based gene therapy in mice diagnosed with pulmonary fibrosis owing to treatment with low doses of the lung-damaging agent bleomycin in the context of short telomeres, a scenario that resembles pulmonary fibrosis in humans associated with short telomeres. Our findings demonstrate that *Tert* treatment significantly improves pulmonary function, decreases inflammation, and accelerates fiber disappearance in fibrotic lungs as early as 3 weeks after viral treatment, resulting in a more rapid improvement or disappearance of the fibrosis. At the molecular level, AAV9-treatment results in telomere elongation and increased proliferation of ATII cells, also significantly decreasing DNA damage, apoptosis, and senescence in these cells. Further supporting these findings, telomerase treatment induces gene expression changes indicative of increased proliferation, lower inflammation and decreased fibrosis in isolated ATII cells.

## Results

### *Tert* targeting of alveolar type II cells prevents pulmonary fibrosis progression induced by short telomeres and restores lung health

Here, we set to address whether telomerase treatment of adult mouse lungs by using AAV9-*Tert* vectors could effectively prevent the progression of pulmonary fibrosis provoked by damage to the lungs (ie.,low-dose bleomycin) and the presence of short telomeres (*Povedano et al., 2015*), a scenario that resembles both familiar and sporadic cases of the human disease (*Alder et al., 2008*; *Armanios, 2013*; *Armanios et al., 2007*). To this end, we used our previously described mouse model of progressive pulmonary fibrosis induced by a low bleomycin dose in the context of short

telomeres (*Povedano et al., 2015*). In particular, owing to the fact that short telomeres per se in the context of the telomerase-deficient mouse model are not sufficient to induce pulmonary fibrosis in mice (*Alder et al., 2011*), we previously generated a mouse model for pulmonary fibrosis associated with short telomeres by treating telomerase-deficient mice from the second (G2) and fourth (G4) generation, G2-G4 *Tert$^{-/-}$* with a low dose of bleomycin (0.5 mg/kg body weight). This low dose of bleomycin is not sufficient to induce pulmonary fibrosis in wild-type mice, but leads to progressive pulmonary fibrosis in the G2-G4 *Tert$^{-/-}$* (*Povedano et al., 2015*). It is relevant to note that this is in contrast to the widely used mouse model of pulmonary fibrosis using a much higher dose of bleomycin (2 mg/kg body weight), which leads to pulmonary fibrosis in wild-type mice but does not recapitulate the short telomere phenotype present in human patients (see *Figure 1—figure supplement 1A*). In particular, we show here that male wild-type mice inoculated either with vehicle or with the standard high-dose bleomycin protocol did not show any significant telomere length changes 4 weeks after bleomycin challenge compared to vehicle inoculated mice (*Figure 1—figure supplement 1A*), suggesting that this mouse model of pulmonary fibrosis does not recapitulate one of the molecular features of the human disease (ie, the presence of short telomeres). Further supporting this notion, treatment of these mice with AAV9-*Tert* did not show any significant decrease in the amount of fibronectin compared to the empty vector-treated lungs (*Figure 1—figure supplement 1B*).

To test the efficacy of telomerase gene therapy in our mouse model of pulmonary fibrosis induced by DNA damage to the lungs (ie.,low-dose bleomycin) and the presence of short telomeres (*Povedano et al., 2015*), we selected AAV9 serotype owing to its high viral transduction of the lungs, and its low immunogenicity (*Bell et al., 2011*; *Zincarelli et al., 2008*). In particular, we previously showed that AAV9-*Tert* transduced lungs cells showed *Tert* mRNA over-expression for at least 8 month post-infection of the vector, as well as resulted in re-activation of telomerase as determined by Telomerase Repeated Amplification Protocol (TRAP) in adult lungs (*Bernardes de Jesus et al., 2012*). To determine the transduction efficiency of the lungs in our current study, we intravenously (IV) injected wild-type adult mice with AAV9-*eGFP* and determined eGFP expression in the lungs 2 weeks later. We found transduction of 3% (GFP positive cells) of total lung cells (*Figure 1A*). Next, we addressed which adult lung cell types were being transduced with the AAV9 vector. We previously described that alveolar type II cells (ATII) cells are a key cell type in the origin of pulmonary fibrosis owing to dysfunctional telomeres (*Povedano et al., 2015*). Thus, we performed double immunofluorescence against eGFP and the surfactant protein C (Sftpc), a specific marker of ATII cells. We first observed that 13.4% of total lung cells were ATII cells (Sftpc-positive cells) (*Figure 1A*), which is in line with the 12–15% reported abundance of ATII cells in whole lung cell population (*Dobbs, 1990*; *Van der Velden et al., 2013*). Importantly, we observed that 17% of total ATII cells were transduced by AAV9-*eGFP* (GFP-positive) (*Figure 1A*). Indeed, more than 80% of all the GFP-positive lung cells were ATII cells (*Figure 1A*), indicating that AAV9 has a specific tropism for these cells.

Next, we treated telomerase-deficient male mice from the second generation, G2 *Tert$^{-/-}$* mice, with the low bleomycin dose (0.5 mg/kg BW) (*Povedano et al., 2015*). Two weeks after bleomycin treatment, we performed computed tomography (CT) to identify those mice with abnormal radiological images of the lungs, indicative of inflammation and pulmonary fibrosis (*Figure 1B*). Approximately 50% of the mice showed an abnormal radiographic pattern presenting reticular opacities suggestive of pulmonary fibrosis (*Povedano et al., 2015*). These mice with an abnormal CT pattern were divided in two random groups, one group was intravenously (IV) injected with AAV9-*Tert* and the other group was injected with the empty vector, as control placebo group. Disease progression was followed longitudinally in both cohorts both by performing weekly spirometry during the first 3 weeks after viral treatment, to measure lung function, as well as by CT imaging at 1, 2, 4 and 7 weeks post viral treatment to follow progression of the abnormal radiographic patterns (*Figure 1B*). Interestingly, only one week after viral treatment, CT imaging showed that all abnormal radiological images in AAV9-*Tert* treated mice regressed in size, while they further increased in size in mice treated with the empty vector (*Figure 1C,D*). After the second week of treatment, we observed a regression of the affected CT lung volume in both groups, although at all time points analyzed the AAV9-*Tert* treated mice showed significantly smaller volume of the CT lesions as compared to mice treated with the empty vector (*Figure 1C,D*). Importantly, at week seven after treatment with the viral vectors (week nine after the induction of fibrosis with bleomycin), the affected CT lung volume

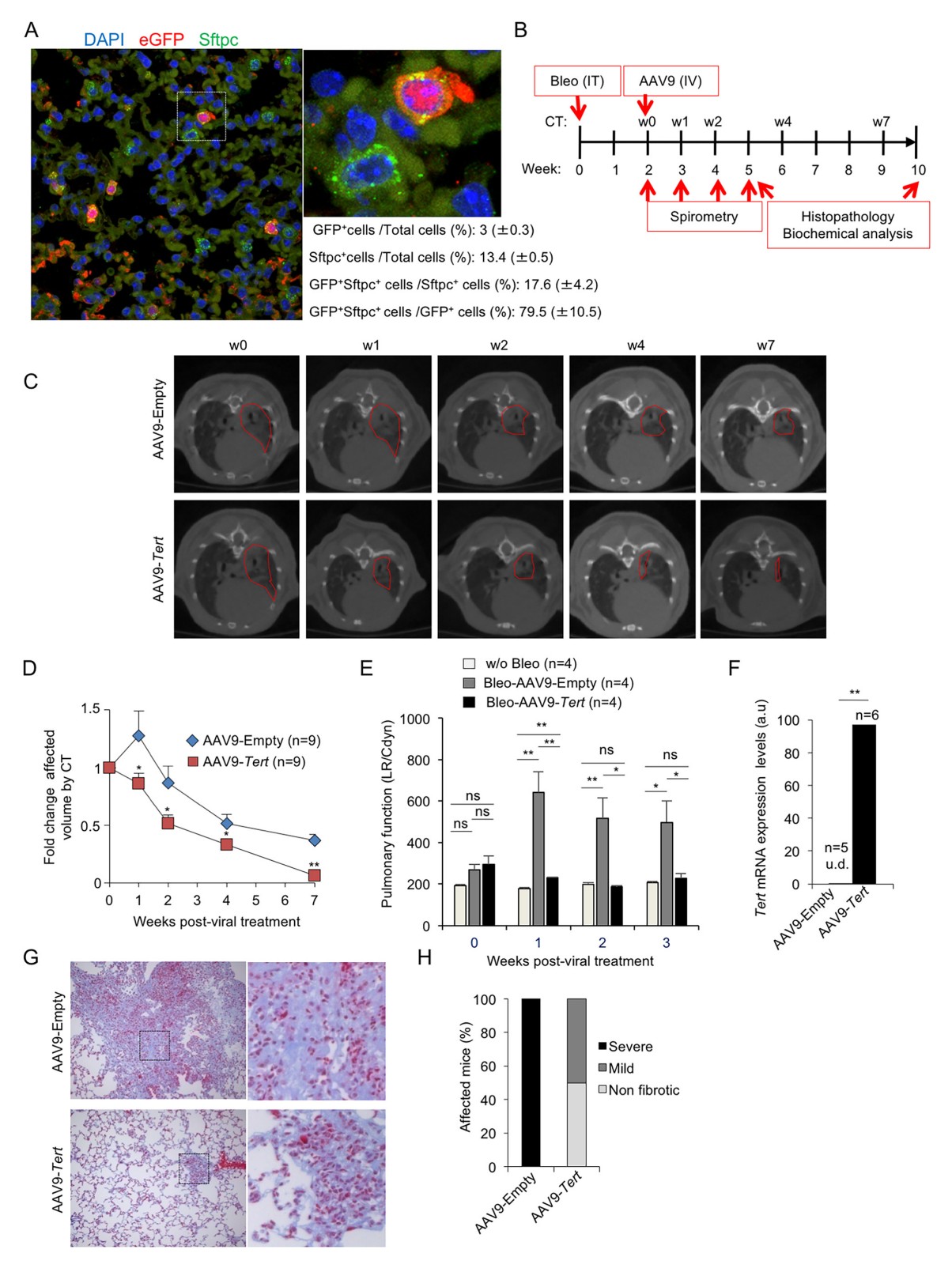

**Figure 1.** AAV9-*Tert* treatment targets ATII cells leading to remission of pulmonary fibrosis. (A) Representative image of immunofluorescence against eGFP (in red) and Sftpc (in green). Mice were injected intravenously in the tail with AAV9-*eGFP* and sacrificed two weeks later to determine virus cell type target. The quantification of percentage of GFP⁺ Sftpc⁺ cells relative to total GFP⁺ cells and to total Sftpc⁺ cells is shown. (B) Eight-ten week old male G2*Tert*⁻/⁻ mice were intratracheally inoculated with 0.5 mg/kg BW bleomycin and two weeks after computed tomography (CT)-diagnosed with

*Figure 1 continued*

pulmonary fibrosis (PF). Affected mice were treated intravenously either with AAV9-empty or AAV9-*Tert*. Spirometric follow–up was performed at 1, 2 and 3 weeks post-viral treatment with the viral vectors. CT follow-up was performed at 1, 2, 4 and 7 weeks post-treatment with the viral vectors. Mice were sacrificed at 3 and 8 weeks post-treatment with the viral vectors for further biochemical and histopathological lung examination. (C) CT representative images for every time point of the treatment (fibrotic area in red). (D) Quantification of fold change affected lung volume with PF normalized to the affected volume before the viral treatment by computed tomography (CT). (E) Follow-up of pulmonary function measured as the ratio between lung resistance and dynamic compliance (LR/Cdyn) (F) *Tert* transcriptional levels in lung 8 weeks post-viral treatment. a.u., arbitrary units (G) Masson´s trichrome staining from lung sections to evaluate fibrotic regions at end point 8 weeks post-viral treatment (collagen fibers in blue; nuclei and erythrocytes in red). (H) Histopathological analysis and fibrosis score from lung sections at end point. The number of mice analyzed per group is indicated. T-test was used in D, E and F and $\chi^2$ analysis in H and I for statistical analysis. *p=0.05; **p<0.01.

DOI: https://doi.org/10.7554/eLife.31299.003

The following figure supplement is available for figure 1:

**Figure supplement 1.** High dose of bleomycin induces pulmonary fibrosis without affecting telomere length in lung cells.

DOI: https://doi.org/10.7554/eLife.31299.004

in the AAV9-*Tert* treated mice corresponds to only 5% of total lung volume, while at this point mice treated with the empty vector still exhibited 40% of total lung volume affected as indicated by CT (*Figure 1C,D*).

Given the small size of mouse lungs, CT imaging as PF diagnose is not fully accurate since inflammation can also give rise to abnormal CT pattern. As an independent longitudinal non-invasive indicator of improvement of pulmonary lesions in the treated mice, pulmonary function was determined by using plethysmography (spirometry) that measures the amount of air left in the lung after deep inhalation and forced exhalation, both previous to viral treatment and during the first 3 weeks after treatment. We observed that lung function measured as the ratio between lung resistance and dynamic compliance worsen in the AAV9-Empty treated mice compared to the AAV9-*Tert* controls already the first week after viral treatment. Importantly, pulmonary function in AAV9-*Tert* treated mice became similar to that of healthy mice non-treated with bleomycin at two weeks post-viral treatment and was maintained thereafter, illustrating the efficacy of the treatment at restoring lung health. In contrast, mice treated with AAV9-Empty show significant higher LR/Cdyn values as compared to healthy and to AAV9-*Tert,* indicating a worsened pulmonary function (*Figure 1E*).

Finally, in order to confirm the areas of the lung affected with fibrosis, at week eight after viral treatment with the vectors, all mice were sacrificed for histopathological, biochemical, and molecular analysis of the lungs. First, we confirmed increased expression of *Tert* mRNA by qPCR in the lungs of AAV9-*Tert* treated mice compared to mice treated with the empty vector, which lacked detectable *Tert* mRNA expression (*Figure 1F*) (*Bär et al., 2014*; *Bernardes de Jesus et al., 2012*).

We next used Masson´s trichrome staining to quantify the lung fibrotic areas by histochemistry. We considered 'severe fibrosis' when more than 30% of the lung parenchyma was affected by fibrosis; 'mild fibrosis' when less than 10% of the lung parenchyma was affected; and 'non-fibrotic lungs' when no signs of fibrosis were found. We found that at week 8 after treatment with the viral vectors (week 10 after the induction of fibrosis), all mice treated with the empty vector showed severe fibrosis as indicated by more than 30% of the lung parenchyma affected by fibrosis (*Figure 1G,H*). In contrast, none of the AAV9-*Tert* treated mice showed severe fibrosis at this point. Instead, 50% of *Tert*-treated mice presented mild fibrosis lesions and 50% were completely free of fibrotic lesions (*Figure 1G–H*). Thus, 50% AAV9-*Tert* treated mice showed undetectable fibrosis as determined by Masson´s trichrome staining at 8 weeks post-viral treatment, while all empty vector treated mice still showed severe fibrotic lesions.

Picrosirius red staining of lungs to determine collagen deposition, confirmed that AAV9-*Tert* treated mice presented one-third less collagen deposition compared to mice treated with the empty vector (*Figure 2A,B*). As an independent biochemical method, we determined collagen peptides containing hydroxyproline on whole lung tissue at 8 weeks after treatment with the viral vectors using liquid chromatography-tandem mass spectrometry (LC-MS/MS) which has been previously validated to quantify collagen (*Chaerkady et al., 2013*; *Montgomery et al., 2012*; *Ono et al., 2009*; *Qiu et al., 2014*; *Taga et al., 2014*). As validation of the technique in our experimental setting, we determined collagen peptides containing hydroxyproline in non-fibrotic lungs from *Tert*-deficient mice that had not been treated with bleomycin and in fibrotic lungs from *Tert*-deficient mice treated

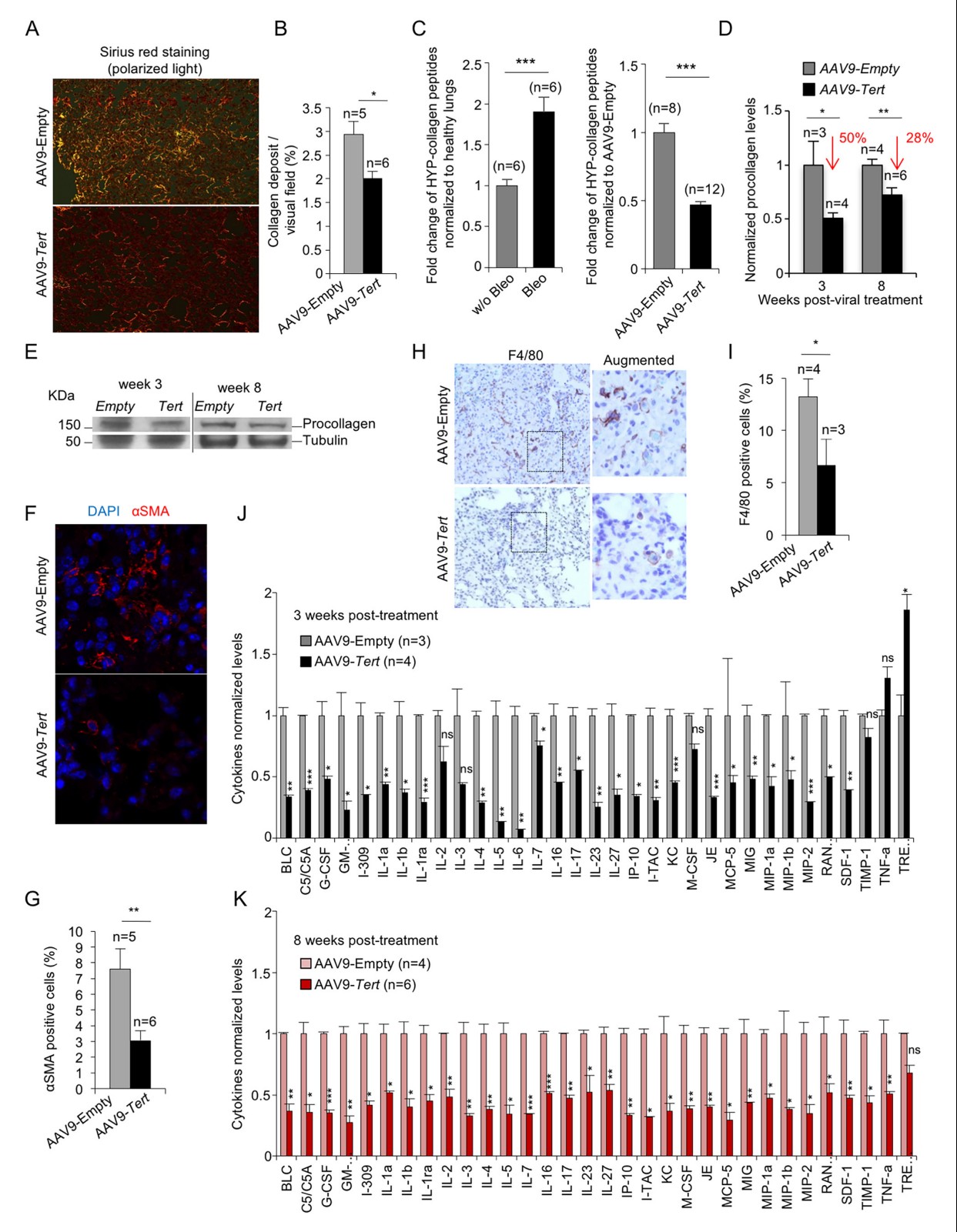

**Figure 2.** AAV9-*Tert* treatment leads to lower collagen deposition, less inflammation and decreased active fibrotic foci. (**A**) Representative images of picosirius red staining visualized by polarized light where collagen fibers are bright orange from mice treated either with AAV9-*Tert* or empty vector 8 weeks post-viral treatment. (**B**) Percent of lung area filled with collagen fibers 8 weeks post-viral treatment. (**C**) Quantification of specific collagen peptides containing hydroxyproline in healthy lungs without bleomycin and in fibrotic lungs 5 weeks after bleomycin insult (left panel) and in lungs

*Figure 2 continued on next page*

*Figure 2 continued*

treated either with *Tert* or empty vector at 8 weeks post-treatment (right panel). (D–E) Quantification of total procollagen levels (D) and representative Western Blot images (E) in lung samples of AVV9-*Tert* and AVV9-empty infected lungs at 3 and 8 weeks post-viral treatment. (F) Representative images of immunofluorescence for αSMA (in red) and DAPI (in blue) at 8 weeks post-viral treatment. (G) Quantification of αSMA positive cells at 8 weeks post-treatment. (H) Representative images of F4/80 (macrophage specific marker) immunohistochemistry staining in AAV9-Empty and AAV9-*Tert* treated mice at 8 weeks post-viral treatment. (I) Quantification of F4/80 positive cells at 8 weeks post-viral treatment. (J–K) Quantification of the indicated cytokines in lung samples of AVV9-*Tert* and AVV9-empty infected lungs at 3 (J) and 8 (K) weeks post-viral treatment. Data represent the mean ±SE of analyzed mice within each group. The number of mice analyzed per group is indicated. T-test was used for statistical analysis. *p=0.05; **p<0.01; ***p<0.001.

DOI: https://doi.org/10.7554/eLife.31299.005

with bleomycin at 5 weeks post-bleomycin treatment (0.5 mg/kg BW) (*Figure 2C*, left panel). The results show that bleomycin treated lungs present a 2-fold increase in the amount of collagen peptides containing hydroxyproline compared to control lungs not treated with bleomycin (w/o bleomycin) in agreement with induction of fibrosis by bleomycin in *Tert*-deficient mice (*Figure 2C*, left panel), thus validating this method for quantification of fibrosis. At 8 weeks after viral treatment (10 weeks post bleomycin treatment), AAV9-*Tert* treated lungs showed 2-fold lower content in collagen peptides containing hydroxyproline compared to AAV9-Empty treated lungs, indicating that telomerase treatment improves collagen removal (*Figure 2C*, right panel). Analysis of total procollagen levels in the lung using western blot analysis also showed approximately 50% and 30% lower levels of procollagen in AAV9-*Tert* treated mice compared to the controls at 3 and 8 weeks post-viral treatment, respectively, suggesting that *Tert* gene therapy leads to a more rapid removal of fiber deposition (*Figure 2D,E*). In line with lower collagen, AAV9-*Tert* treated lungs also showed significantly less αSMA-positive myofibroblasts compared to empty vector-treated mice, in agreement with the fact that these cells are associated with collagen deposition in human IPF patients (*Figure 2F,G*), thus suggesting an inactivation of fibrotic foci upon *Tert* treatment.

Finally, also in agreement with fibrosis regression and tissue healing in mice treated with telomerase, AAV9-*Tert* treated mice showed significantly less macrophage infiltrates as detected by F4/80 staining in their remaining fibrotic areas compared with AAV9-Empty treated mice (*Figure 2H,I*), suggestive of decreased inflammation. We confirmed lower inflammation in the AAV9-*Tert* treated mice compared to the empty vector treated group by quantification of a large panel of cytokines including BLC, C5/C5A, G-CSF, I-309, IL-1a, IL-1b, IL-1ra, IL-2, IL-3, IL-4, IL-5, IL-6, IL-7, IL-16, IL-17, IL-23, IL-27, IP-10, I-TAC, KC, M-CSF, JE, MCP-5, MIG, MIP-1a, MIP-1b, MIP-2, RANTES, SDF-1, TIMP-1, TNF-a and TREM-1 by Elisa. In particular, already at 3 weeks after viral treatment, AAV9-*Tert* treated mice showed significantly lower levels of these cytokines compared to the cohort treated with the empty vector and this was maintained at 8 weeks after viral treatment, indicating the efficacy of the therapy in dampening the inflammatory response (*Figure 2J,K*). Thus, these results demonstrate lower inflammation in the lungs of *Tert* treated mice compared to the controls.

## AAV9-*Tert* treatment rescues apoptosis and cellular senescence in fibrotic lungs

Short dysfunctional telomeres have been previously shown to trigger a persistent DNA damage response (DDR) characterized by increased γH2AX foci, increased expression of *p21* and *p53* cell arrest and senescence markers, as well as induction of apoptosis (*Hemann et al., 2000*; *Meier et al., 2007*). Indeed, we previously described that mice with pulmonary fibrosis owing to short telomeres also show increased γH2AX foci, increased expression of *p21* and *p53* senescence markers, as well as induction of apoptosis in the lungs (*Povedano et al., 2015*). Interestingly, analysis of these molecular markers in the lungs of treated mice at two time points after viral treatment shows that the lungs of AAV9-*Tert* treated mice have a significant reduction of DNA damage already at 3 weeks post-viral treatment which is also maintained at 8 weeks post-viral treatment (endpoint of the experiment), as indicated by lower percentage of cells positive for γH2AX compared to mice treated with empty vector (*Figure 3A,B*). Consistently, we also found a significant reduction in the abundance of p21 and p53-positive cells in AAV9-*Tert* treated mice compared to those treated with the empty vector as early as 3 weeks after viral treatment and again these lower levels were maintained at 8 weeks post-viral treatment (*Figure 3A,B*). Moreover, we also observed a significant decrease in caspase 3-

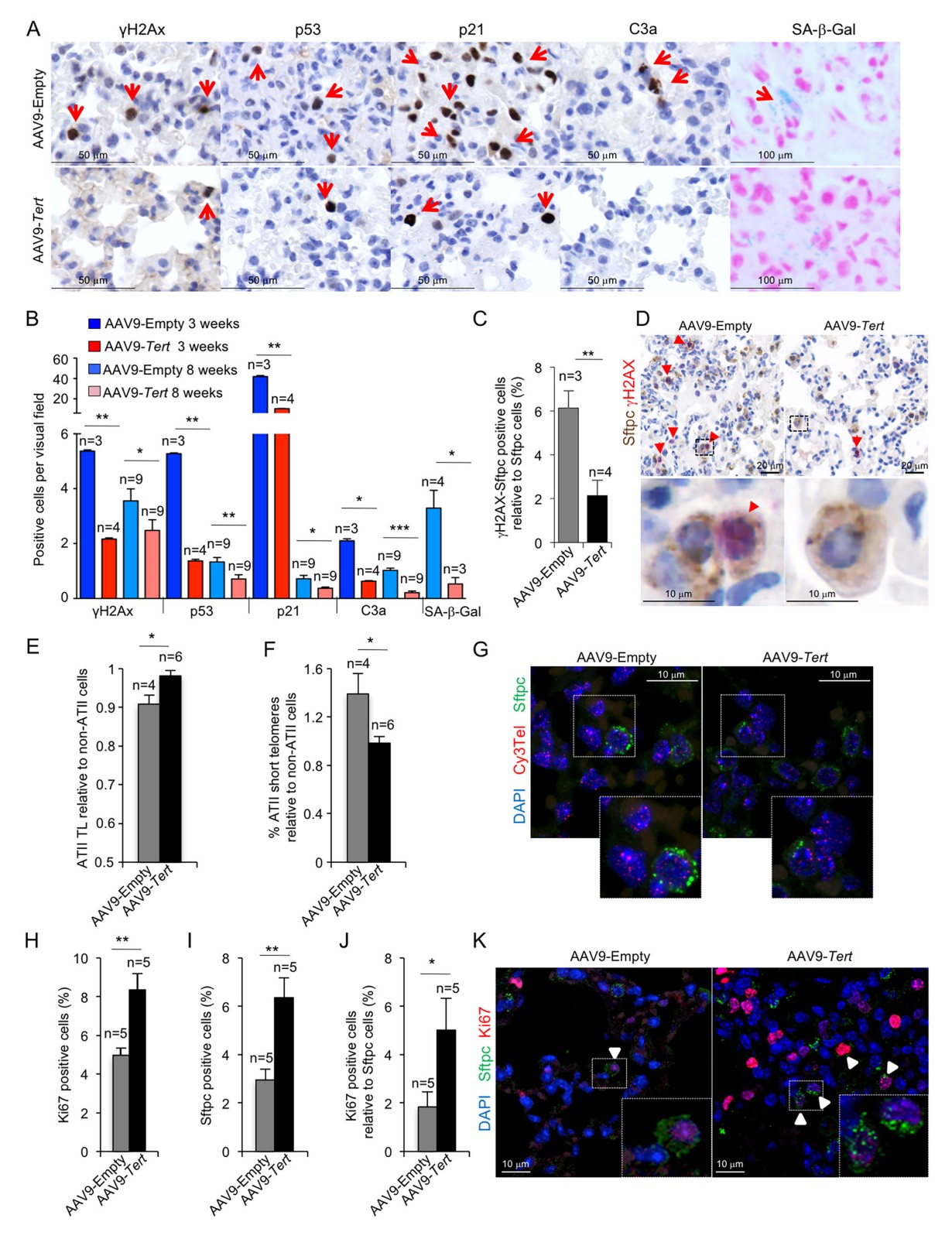

**Figure 3.** AAV9-*Tert* treatment reduces DNA damage, improves telomere maintenance and proliferation in ATII cells.  (**A**) Representative images for γH2AX, p53, p21, active caspase 3 (C3a) and SA-β-Gal stained lungs at 3 weeks post-viral treatment. (**B**) Quantification of γH2AX, p53, p21, C3a and SA-β-Gal positive cells per visual field of lungs treated either with *Tert* or empty vector at 3 and 8 weeks post-viral treatment. (**C**) percentage of damaged (γH2AX positive) ATII cells (Stfpc positive) in lungs treated either with *Tert* or empty vector at three post-viral treatment. (**D**) Representative images of
*Figure 3 continued on next page*

*Figure 3 continued*

double immunohistochemistry against Stfpc (brown) and γH2AX (red) of lungs treated either with *Tert* or empty vector at three post-viral treatment. (E–F) Fold change in telomere length (E) and percentage of short telomeres (F) in ATII cells relative to non-ATII cells at 8 weeks post-viral treatment. (G) Representative images of immuno-QFISH with Cy3Telomere probe (in red), Sftpc (in green) and DAPI (in blue) in lungs at 8 weeks post-viral treatment. (H–J) Quantification of percentage of Ki67 positive cells (H), Sftpc positive cells (I) and Ki67 positive cells relative to Sftpc positive cells at 8 weeks post-viral treatment (J). (K) Representative images of double immunofluorescence against Sftpc (in green) and Ki67 (in red) in lungs at 8 weeks post-viral treatment. Data represent the mean ±SE of analyzed mice within each group. The number of mice analyzed per group is indicated. T-test was used for statistical analysis. *p=0.05; **p<0.01; ***p<0.001.

DOI: https://doi.org/10.7554/eLife.31299.006

positive cells at both time points (3 and 8 weeks post-viral treatment) in the lungs of *Tert*-treated mice compared to the empty-treated cohort (*Figure 3A,B*), indicative of decreased apoptosis. Together, these results indicate that *Tert* expression in the lungs of adult mice with pulmonary fibrosis is sufficient to decrease DNA damage and apoptosis, as well as to decrease the levels of p21 and p53, as early as 3 weeks after viral treatment and this is maintained until the end-point of the experiment at 8 weeks post-viral treatment when the fibrosis was reverted or cured in a significant proportion of mice treated with telomerase.

In order to specifically address presence of senescent cells in the lungs, we performed whole mount staining for SA-β-galactosidase assay in mice diagnosed with pulmonary fibrosis and treated with either AAV9-*Tert* or -Empty vectors. While senescence epithelial cells were readily detected in mice diagnosed with pulmonary fibrosis and treated with the empty vector, they were undetectable in the residual fibrotic areas present in few AAV9-*Tert* treated lungs at the end of the experiment (*Figure 3A,B*). Of note, macrophages and fibroblasts were discarded from the analysis based on cell morphology. These findings indicate that *Tert* gene therapy rescues DNA damage, apoptosis and cellular senescence in mice diagnosed with pulmonary fibrosis owing to critically short telomeres. However, as these histological analyses do not permit distinguishing among different cell types, to specifically address the DNA damage burden in ATII cells we performed double immunohistochemistry staining with anti-SFTPC to mark ATII cells and anti-γH2AX to mark cells with DNA damage (*Figure 3C,D*). The results clearly show that the amount of damaged ATII cells in AAV9-*Tert* treated lungs is reduced by 3-fold compared to control mice treated with the AAV9-Empty vector at 3 weeks post-viral treatment (*Figure 3C,D*).

## AAV9-*Tert* treatment results in increased proliferation of ATII cells

To further understand the molecular mechanisms by which *Tert* gene therapy results in significant remission and healing of pulmonary fibrosis owing to short telomeres, we next studied telomere length specifically in the ATII cells of mice treated with either AAV9-*Tert* or the empty vector. To this end, we performed an Immuno-FISH using a telomeric PNA probe and a Sftpc antibody to specifically mark ATII cells in lung samples at 8 weeks post-viral treatment. We analyzed telomere intensity in both Sftpc positive (ATII) and negative (non-ATII) cells. We found that ATII cells have shorter telomeres than non-ATII cells in telomerase-deficient mice in agreement with our previous findings indicating that these cells are important for the regeneration of lung damage induced by dysfunctional telomeres, as they have undergone more cell divisions (*Povedano et al., 2015*). Interestingly, ATII cells from mice treated with AAV9-*Tert* showed the same telomere length than the surrounding non-ATII cells, suggesting that telomerase treatment is preserving telomeres in these cells in the context of lung fibrosis (*Figure 3E–G*). Indeed, the percentage of short telomeres of ATII cells compared to non-ATII in empty vector-treated mice (ATII/non-ATII ratio = 1.4) is significantly higher than in AAV9-*Tert* treated mice (ATII/non-ATII ratio = 0.98). We considered short telomeres those spots with an intensity ≤30 a.u. corresponding to 20th percentile. The percentage of short telomeres from ATII cells was normalized to non-ATII cells to avoid inter-individual variability (*Figure 3E,G*). In summary, these results indicate that specific telomerase targeting to ATII cells results in improved telomere length maintenance and a consequent reduction in DNA damage burden of these cells compared to ATII cells from mice treated with the empty vector.

Next, we addressed the effects of *Tert* treatment on the ability of ATII cells to proliferate and regenerate the damaged lung tissue upon diagnosis of fibrosis. To this end, double immunofluorescence against the ATII cells-specific marker Sftpc and the proliferation marker Ki67 was performed in

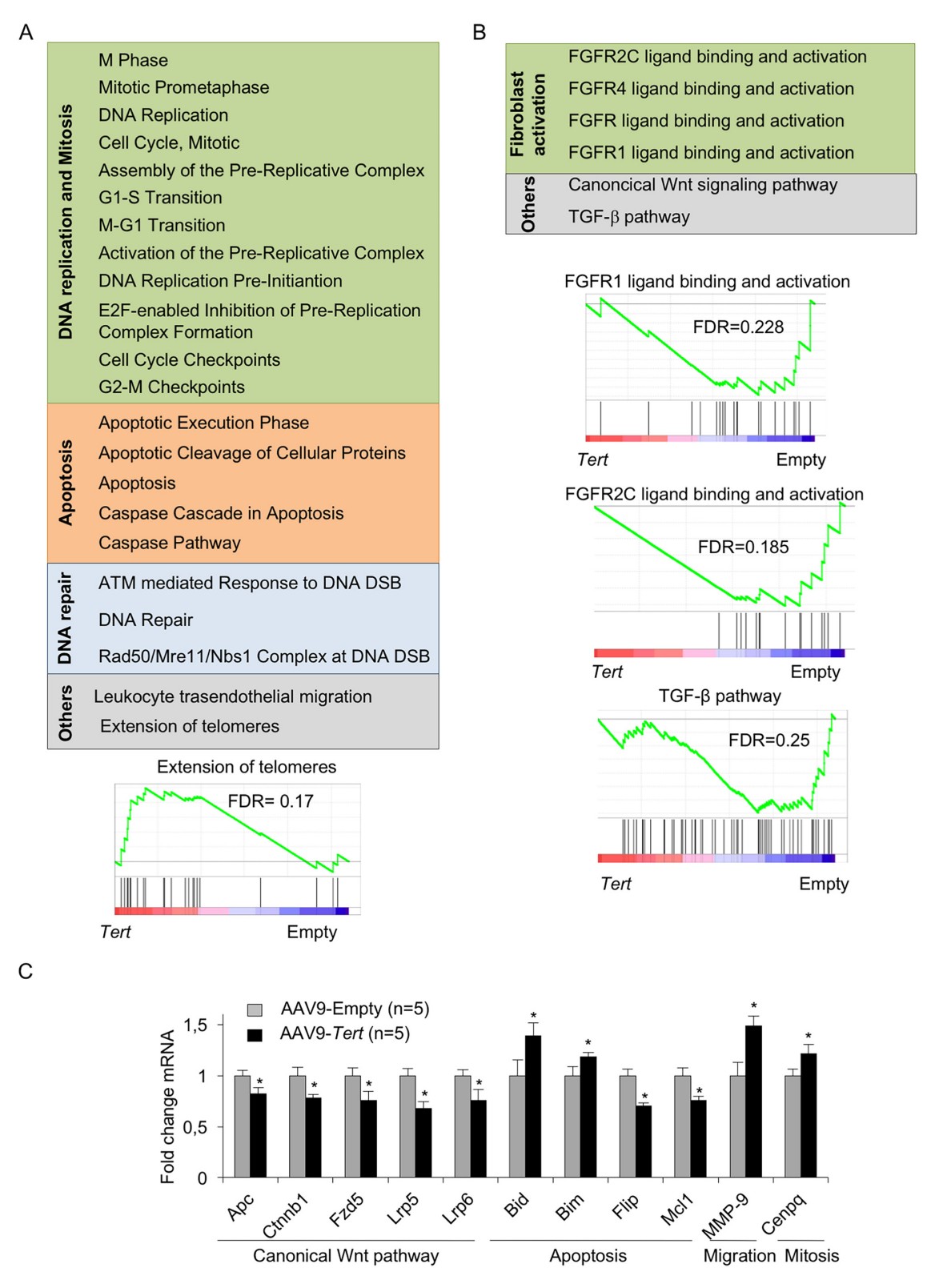

**Figure 4.** AAV9-*Tert* treatment induces transcriptional changes in the lungs. (**A–B**) Summary table indicating various significantly (FDR < 0.25) upregulated (**A**) and downregulated (**B**) pathways in AAV9-*Tert* compared with AAV9-Empty treated lungs at 8 weeks post-viral treatment. Examples of GSEA plots for the indicated pathways are shown below. Microarray genes were ranked based on the two-tailed t-statistic tests obtained from the AAV9-*Tert* versus AAV9-Empty by pair-wise comparisons. The red to blue horizontal bar represents the ranked list. Those genes showing higher

*Figure 4 continued on next page*

*Figure 4 continued*

expression levels for each cohort are located at the edges of the bar (AAV9-Empty; AAV9-*Tert*). The genes located at the central area of the bar show small differences in gene expression fold changes between both groups. (C) Fold change mRNA expression levels of candidate genes related with canonical Wnt pathway, apoptosis, mitosis and transendothelial migration in AAV9-*Tert* relative to empty vector. For GSEA Kolmogorov–Smirnoff testing was used for statistical analysis. The FDR is calculated by Benjamini and Hochberg FDR correction. Data represent the mean ± SE of analyzed mice within each group. The number of mice analyzed per group is indicated. T-test was used for q-PCR statistical analysis. *p=0.05.

DOI: https://doi.org/10.7554/eLife.31299.007

The following figure supplements are available for figure 4:

**Figure supplement 1.** Differentially expressed genes from AAV9-*Tert* treated mice correlate with ATII cells gene expression signature.

DOI: https://doi.org/10.7554/eLife.31299.008

**Figure supplement 2.** *Tert* overexpression in fibrotic lungs mimic neonatal regenerative heart tissue after infarctation.

DOI: https://doi.org/10.7554/eLife.31299.009

lung samples at 8 weeks post-viral treatment. We found that *Tert* treated mice showed a 2-fold increase in Ki67 positive cells in whole lung tissue compared to controls (*Figure 3H*). When specifically looking at ATII cells, we observed a 2-fold increase in the total number of Sftpc positive cells and 2.5-fold increase in Ki67 positive ATII cells in AAV9-*Tert* treated lungs compared to the empty vector controls (*Figure 3I–K*). Although we cannot distinguish between the AAV9-*Tert* infected and non-infected ATII cells, the fact that 80% of the total AAV9-infected lung cells are ATII cells (*Figure 1A*), suggests that AAV9-*Tert* treatment is resulting in increased proliferation of these cells leading to a higher potential for lung regeneration and the remission of lung fibrosis. Thus, the higher number of proliferating ATII cells are in agreement with the significant lower percentage of short telomeres in ATII cells in AAV9-*Tert* treated lungs as well as with the significant decrease in the number of p21 and p53 positive cells in AAV9-*Tert* treated lungs (*Figure 3B–J*).

## AAV9-*Tert* treatment leads to gene expression changes indicative of higher regeneration potential

Next, we studied gene expression changes induced by *Tert* expression in the context of lung fibrosis. To this aim, we first performed DNA microarray analysis from the post-caval lung lobe from mice diagnosed with pulmonary fibrosis which were treated either with AAV9-*Tert* or with the empty vector at 8 weeks post viral treatment with the vectors (5 mice were included per group). We found that only 53 genes were significantly upregulated (False Discovery Rate, FDR < 0.05) in AAV9-*Tert* treated mice compared to mice treated with the empty vector (*Supplementary file 1*). This FDR cut-off highlights the significance of the gene expression changes observed, as only 5% of the hits are expected to be false positive. Gene set enrichment analysis (GSEA) showed significantly deregulated pathways between both groups (*Figure 4A–B*). Those pathways found upregulated in AAV9-*Tert* treated lungs presented a signature related with DNA replication and mitosis, apoptosis, DNA repair, the leukocyte transendothelial migration pathway, and extension of telomeres (*Figure 4A*, *Figure 4—figure supplement 1A*). Upregulation of the 'extension of telomeres pathway' is in line with improved telomere maintenance in ATII cells treated with *Tert* (*Figure 4A*). Similarly, upregulation of DNA replication and mitosis pathways is in line with increased proliferation in ATII cells (*Figure 4A*, *Figure 4—figure supplement 1A*). In contrast, pathways downregulated in AAV9-*Tert* compared to AAV9-Empty treated lungs were related to fibroblast growth factor receptors, Wnt and TGF-β pathways (*Figure 4B*, *Figure 4—figure supplement 1A*). Of interest, four of the pathways downregulated by *Tert* are related with fibroblast growth factor receptors; that is, the FGFR2C, FGFR4, FGFR and the FGFR1 ligand binding and activation cascades (*Figure 4B*). As FGF1-FGFRc over-expression has been shown to contribute to pathogenesis in IPF patients (*MacKenzie et al., 2015*), these findings suggest that *Tert* impairs fibroblast activation, thus facilitating fibrosis regression. In line with this, we also found a downregulation of the TGF-β pathway in the AAV9-*Tert* treated lungs. This is in agreement with our previous findings that *Tert* overexpression in mouse embryonic fibroblasts (MEFs) induces downregulation of TGF-β (*Geserick et al., 2006*). TGF-β pathway has been linked to fibroblast activation and differentiation to myofibroblast (*Pedroza et al., 2016*). Indeed, pirfenidone, a drug that blocks TGF-β can significantly slow pulmonary fibrosis progression (*Hunninghake, 2014*; *Karimi-Shah and Chowdhury, 2015*; *King et al., 2014*).

By using qRT-PCR, we validated a random selection of the more differentially expressed genes within these pathways: *Apc*, *Ctnnb1*, *Fzd5*, *Lrp6*, *Lrp5*, *Bim*, *Flir*, *Bid*, *Mcl1*, *Mmp-9* and *Cenpq* (*Figure 4C*). Of note, downregulation of the Wnt pathway in AAV9-*Tert* treated adult lungs is in contrast with the notion that TERT can activate Wnt/β-catenin pathway during development (*Park et al., 2009*). Instead, our results go in line with recent findings showing that high levels of the Wnt pathway genes *Lrp5* and *Lrp6* are linked to bad prognosis for IPF patients (*Lam et al., 2014*). Interestingly, we found *Mmp9* upregulation in *Tert* treated lungs, in line with the fact that *Mmp9* overexpression attenuates fibrosis in bleomycin-induced IPF (*Cabrera et al., 2007*).

Next, we set to analyze whether the observed gene expression changes corresponded to lung epithelial cells. To this end, we compared the AAV9-*Tert* lung signature with the genes normally expressed in different lung populations (*The Gene Expression Barcode 3.0*). Most upregulated genes

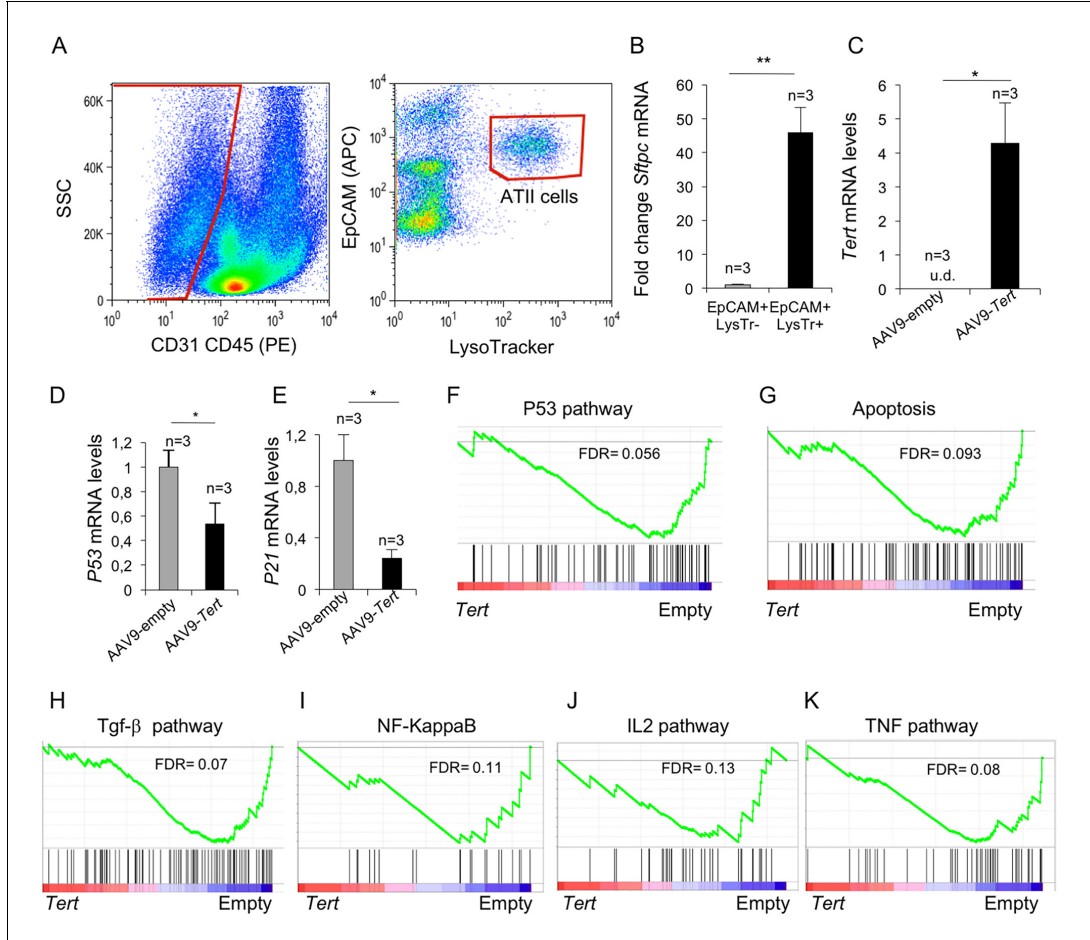

**Figure 5.** Isolated ATII cells overexpress *Tert* and show downregulation of DDR- and inflammatory- related pathway. (A) FACs representative dot plots of lungs one week post either AAV9-Empty or AAV9-*Tert* treatment. The epithelial cell population was identified as CD31/CD45 double negative. ATII cells were identified by LysoTracker and EpCAM doubly positive cells and isolated by cell sorting. (B) Validation of specific ATII cells marker *Sftpc* by RT-qPCR. (C) Transcriptional levels of *Tert* in isolated ATII cells from lungs treated with the indicated vectors. (D–E) mRNA expression levels of *p53* (C) and *p21* (D) genes in ATII by RT-qPCR from lungs treated with the indicated vectors. (F–G) GSEA plots for the indicated downregulated DDR related pathways in AAV9-*Tert* infected ATII cells. (H–K) GSEA plots for the indicated downregulated inflammatory related pathways in AAV9-*Tert* infected ATII cells. Microarray genes were ranked based on the two-tailed t-statistic tests obtained from the AAV9-*Tert* versus AAV9-Empty by pair-wise comparisons. The red to blue horizontal bar represents the ranked list. Those genes showing higher expression levels for each cohort are located at the edges of the bar (AAV9-Empty; AAV9-*Tert*). The genes located at the central area of the bar show small differences in gene expression fold changes between both groups. Data represent the mean ±SE of analyzed mice within each group. The number of mice analyzed per group is indicated. T-test was used for RT-qPCR statistical analysis. *p=0.05; **p<0.01. For GSEA Kolmogorov–Smirnoff testing was used for statistical analysis. The FDR is calculated by Benjamini and Hochberg FDR correction.

DOI: https://doi.org/10.7554/eLife.31299.010

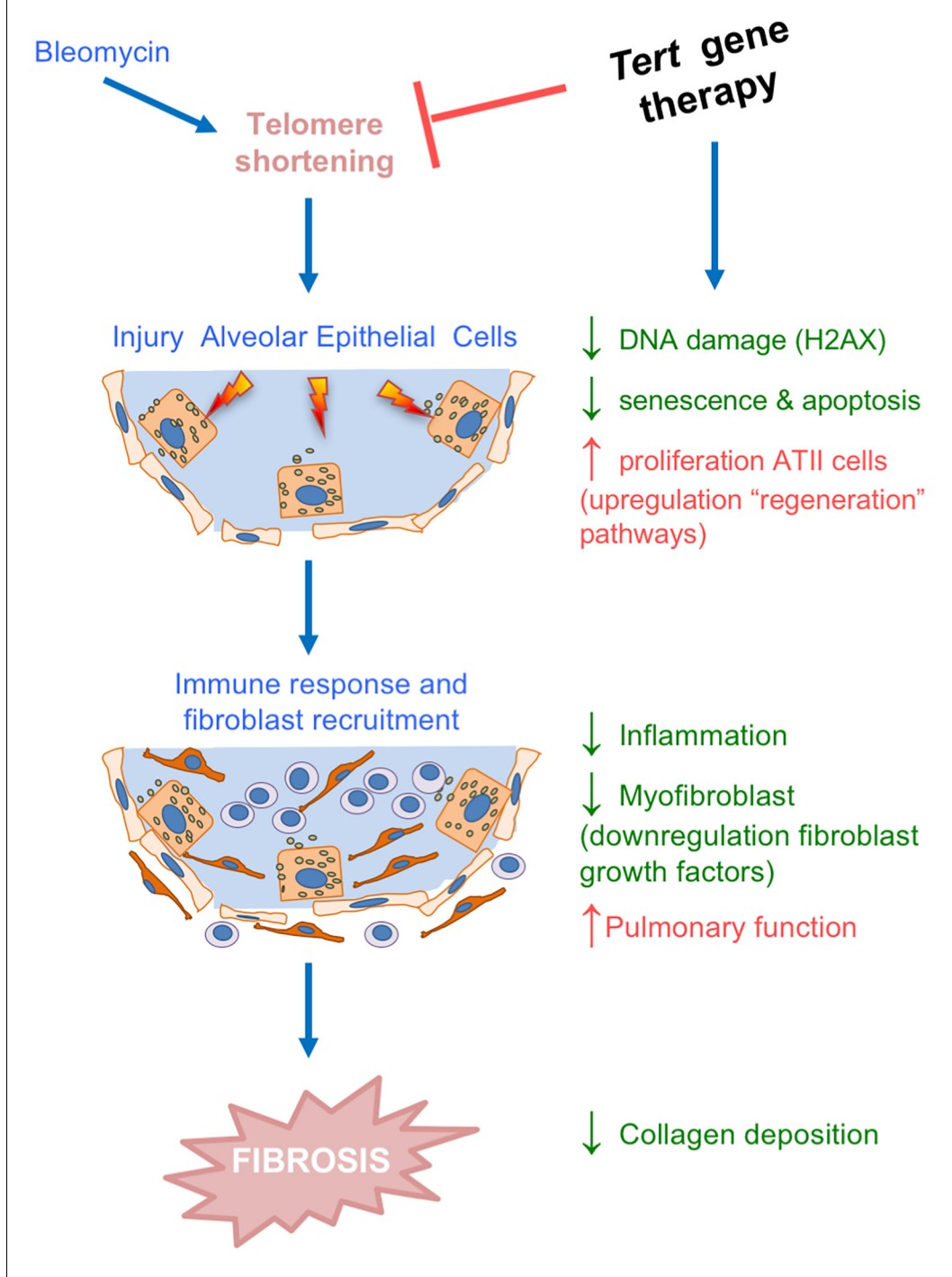

**Figure 6.** *Tert* gene therapy targets the basis of pulmonary fibrosis. Proposed model for the mechanism underlying *Tert* gene therapy. AAV9-*Tert* therapy targets one of the molecular causes of the disease, short telomeres (*Alder et al., 2008*; *Armanios et al., 2007*; *Povedano et al., 2015*), resulting in decreased DNA damage, senescence/apoptosis and improved proliferative potential of the ATII cells and subsequently decreasing inflammation and fibrosis.

DOI: https://doi.org/10.7554/eLife.31299.011

(0 < Fc < 1) corresponded to genes specifically expressed by ATII cells (*Figure 4—figure supplement 1B*), with a minority of the genes being normally expressed in leukocytes or embryonic

fibroblasts. Similar findings were found for the downregulated genes ($-1 < Fc < 0$) (*Figure 4—figure supplement 1C*).

Finally, we find of interest the fact that similar findings were found by us on the amelioration of heart function after infarct in mice treated with AAV9-*Tert* (*Bär et al., 2014*). In particular, *Tert* treatment lead to lower fibrotic scarring of the heart and increased cardiac myocyte proliferation concomitant with transcriptional changes suggestive of a regenerative signature (*Bär et al., 2014*). Interestingly, the gene expression changes in AAV9-*Tert* treated lungs correlate with the regenerative heart signature described in neonatal mice (*Haubner et al., 2012*) as well as those reported by us in the context of improved cardiac regeneration upon infarct by AAV9-*Tert* treatment (*Bär et al., 2014*) (*Figure 4—figure supplement 2*).

To specifically address the gene expression changes stemmed from *Tert* upregulation in ATII cells, we isolated ATII cells at one week after treatment of fibrotic lungs with AAV9-*Tert* and performed transcriptional profiling. ATII cells were identified as EpCAM⁺ LysoTracker⁺ cells and non-ATII cells as EpCAM⁺ LysoTracker⁻ (*Figure 5A*), and expression of the ATII-specific marker *Sftpc* by RT-PCR was used to validate the FACS sorting (*Figure 5B*). FACS-sorted ATII cells from AAV9-*Tert* treated mice showed *Tert* mRNA expression while it was undetectable in FACS-sorted ATII cells from empty vector-treated controls (*Figure 5C*). We also validated decreased *p53* and *p21* mRNA expression by RT-PCR in ATII cells from *Tert* treated mice compared with empty vector treated mice (*Figure 5D,E*), in agreement with lower senescence and DNA damage in *Tert* treated mice (see *Figure 3A,B*).

Importantly, upon gene expression analysis of isolated ATII cells from *Tert*-treated mice, GSEA analysis showed downregulation of p53 signaling and apoptotic pathways (*Figure 5F,G*), consistent with lower DNA damage in lungs from *Tert*-treated mice compared to those from empty vector-treated mice (see *Figure 3A,B*). Also in line with lower fibrosis in the lungs from *Tert*- treated mice, ATII cells showed downregulation of several inflammation related pathways including the TGF-β, NF-KappaB, IL2 and TNF signaling pathways (*Figure 5H–K*). Together, these results further demonstrate that ATII cells are transduced by AAV9-*Tert*, leading to increased telomerase expression, as well as to downregulation of DNA damage and fibrotic pathways.

## Discussion

In spite of recent therapeutic advances for the treatment of pulmonary fibrosis, most patients still face a fatal outcome, where the only curative treatment is lung transplantation. As an example, the recently FDA approved drugs nintedanib and pirfenidone can significantly reduce the progression of pulmonary fibrosis in patients although no full-remissions have been observed (*Hunninghake, 2014*; *Karimi-Shah and Chowdhury, 2015*; *King et al., 2014*).Thus, new therapeutic strategies aimed to cure the disease are still needed.

As short telomeres have been shown to be at the origin of both sporadic and familial cases of pulmonary fibrosis (*Alder et al., 2008*; *Armanios et al., 2007*; *Povedano et al., 2015*), here, we set out to address the potential of telomerase gene therapy in the treatment of these cases of idiopathic pulmonary fibrosis (IPF). To this end, we used a pre-clinical mouse model of pulmonary fibrosis induced by damage to the lungs (ie., treatment with a low bleomycin dose) and the presence of short telomeres (*Povedano et al., 2015*), a scenario that resembles both familiar and sporadic cases of the human disease which are associated with the presence of short telomeres (*Alder et al., 2008*; *Armanios et al., 2007*). It is important to note that only the presence of short telomeres per se in mice deficient for telomerase does not lead to pulmonary fibrosis (PF) (*Alder et al., 2011*). We generated a new, more 'humanized' mouse model for pulmonary fibrosis by subjecting *Tert*⁻/⁻ mice with short telomeres to small doses of bleomycin (0.5 mg/kg body weight). This low dose of bleomycin is not sufficient to induce pulmonary fibrosis in wild-type mice, but synergizes with short telomeres in the context of *Tert*⁻/⁻ mice leading to full-blown, progressive pulmonary fibrosis, recapitulating many of the features of the human disease, including the presence of short telomeres (*Blasco et al., 1997*). We further show here that the most widely used mouse model of pulmonary fibrosis, which is based on treating wild-type mice with a high dose of bleomycin (*Adamson and Bowden, 1974*), do not present short telomeres in the lung. Thus, to our knowledge the mouse model used here is the only available mouse model to date that develops pulmonary fibrosis as a consequence of telomere

length defects as it is also the case of human patients with pulmonary fibrosis associated to short telomeres (**Alder et al., 2008**; **Armanios et al., 2007**).

Here we extensively demonstrate by using biochemical, functional, and histochemistry analysis, that *Tert* gene therapy (AAV9-*Tert*) in mice diagnosed with pulmonary fibrosis leads to a more rapid regression of pulmonary fibrosis and improves pulmonary function as early as weeks 1 and 3 after treatment and this is maintained at the end-point of the experiment at week 8, when a significant percentage of mice show curation of the fibrosis.

Interestingly, we found that the major lung cell type transduced by AAV9 are ATII cells, previously shown by us to be at the origin of pulmonary fibrosis owing to dysfunctional telomeres (**Povedano et al., 2015**). ATII cells have been also proposed to be involved in lung regeneration upon injuries (**Serrano-Mollar et al., 2007**). *Tert* increased expression in ATII cells results in improved telomere maintenance and proliferation of these cells, which is concomitant with lower DNA damage as well as decreased presence of apoptotic and senescence cells already at week 3 after AAV9-*Tert* treatment. As a consequence, *Tert*-treated mice show a better pulmonary function as well as decreased inflammation and decreased fibrosis (lower collagen depots). These results are in line with the notion that short telomeres can impair the ability of stem cells to regenerate tissues (**Blasco, 2007**; **Flores et al., 2008**), and with recent findings suggesting that IPF is the result of defective regeneration upon repetitive epithelial cell injury (**Hinz et al., 2007**; **Ryu et al., 2014**). Of clinical relevance is the observation that these beneficial effects of *Tert* gene therapy are achieved with a transduction efficiency of 3% of total lung cells and 17% of ATII cells.

Of relevance, we show here that *Tert* expression in fibrotic lungs leads to downregulation of pathways involved in fibroblast activation and, in particular, of the TGF-β pathway. Indeed, gene expression analysis of isolated ATII cells from *Tert*-treated mice showed downregulation of p53 signaling, apoptotic pathways and of several inflammation-related pathways including the TGF-β, NF-KappaB, IL2 and TNF signaling pathways as early as 1 week after *Tert* treatment. Dampening of inflammation upon *Tert* treatment was further demonstrated by decreased levels of a large number of cytokines already at 3 weeks post-viral treatment that were maintained all throughout the experiment. These pathways are known to be important players in IPF, and are targeted by the currently approved treatments for this disease, such as pirfenidone (**Inomata et al., 2014**; **Oku et al., 2008**).

Importantly, in contrast to the available IPF treatments, pirfenidone and nintedanib, which are not able to induce disease remission neither in patients nor in preclinical mouse models (**Inomata et al., 2014**; **Oku et al., 2008**; **Tanaka et al., 2012**), we show here that AAV9-*Tert* therapy effectively accelerates the regression of pulmonary fibrosis in mice. We would like to propose that this might be due to the fact that AAV9-*Tert* therapy targets one of the molecular causes of the disease, namely short telomeres (**Alder et al., 2008**; **Armanios et al., 2007**; **Povedano et al., 2015**), which in turn we show here that results in decreased DNA damage and improved proliferative potential of the ATII cells, and subsequently in decreased fibrosis and inflammation (**Figure 6**). In agreement with this, it was shown that treatment with GRN510, a small molecule activator of telomerase, suppresses the development of fibrosis and accumulation of senescent cells in the lung in a model of bleomycin-induced fibrosis (**Le Saux et al., 2013**). In contrast, pirfenidone and nintedanib might be acting on downstream events, particularly on reducing fibrosis, while molecular damage at the origin of the disease (ie. damaged telomeres), as well as the subsequent impairment of the regenerative potential of epithelial cells persists (**Alder et al., 2008**; **Armanios et al., 2007**; **Povedano et al., 2015**). Future ATII lineage tracing experiments would be of interest to ultimately demonstrate that defective regenerative potential of ATII associated to short telomeres is a key molecular event in PF development.

As a note of caution regarding the use of AAV vectors, it has been shown that transcriptional active host loci and DNA repair factors impact on rAAV vector integration in the host genome as well as on vector maintenance as linear or circular episomes, affecting thereby the duration of expression and mutagenic potential of the vector (**Inagaki et al., 2007**; **Nakai et al., 2003**; **Song et al., 2001**; **Song et al., 2004**). Thus, further work is needed to address the potential effects of the PF disease on vector genome processing. In addition, although we have not observed increased cancer incidence by systemic administration of the AAV9-*Tert* vector in different mouse models previously studied in the lab, such as *AAV9-Tert* treatment to delay organismal aging and increase longevity (**Bernardes de Jesus et al., 2012**), *AAV9-Tert* treatment in mouse models of heart infarct (**Bär et al., 2014**), and *AAV9-Tert* treatment in mouse models of aplastic anemia

(*Bar et al., 2016*), further work is needed to address its potential tumorigenic effects in cancer prone scenarios, or in the context of severely damaged tissues, where senescence and apoptosis may be acting as tumor suppressive mechanisms.

In summary, the findings described here demonstrate the therapeutic clinical potential of *Tert* gene therapy to efficiently improve pulmonary fibrosis associated with short telomeres.

# Materials and methods

## Key resources table

| Reagent type (species) or resource | Designation | Source or reference | Identifiers | Additional information |
|---|---|---|---|---|
| gene (Mus musculus) | Tert | NA | Gene ID: 21752 | *Liu et al. (2000)* |
| strain, strain background (Mus musculus) | G2 Tert-/- ; male | NA | | |
| strain, strain background (AAV9) | AAV9-Tert | Other | | *Bernardes de Jesus et al. (2012)* |
| strain, strain background (AAV9) | AAV9-EGFP | Other | | *Bernardes de Jesus et al. (2012)* |
| antibody | anti-p53 | CNIO histopathology core unit | POE316A/E9 | (1:200) |
| antibody | anti-p21 | CNIO histopathology core unit | HUGO-291H/B5 | (1:200) |
| antibody | anti-phospho–H2AX(Ser139) | Merck Millipore | Clone JBW301 | (1:200) |
| antibody | anti-F4/80 | ABD serotec | CI:A3-1 | (1:200) |
| antibody | anti-p19 | Santa Cruz Biotecchnology | 5-C3-1 | (1:50) |
| antibody | anti-Sftpc | Merck Millipore | AB3786 | (1:200) |
| antibody | anti-cleaved Caspase 3 | R& and D systems | | (1:1000) |
| antibody | anti-Sftpc | Santa Cruz Biotecchnology | C-19 | (1:50) |
| antibody | anti-alfaSMA | Biocare Medical | CME 305 | (1:200) |
| antibody | anti-GFP | Roche | 05-636 | (1:100) |
| antibody | anti-Ki67 | Master Diagnostica | 0003110QD | (1:500) |
| antibody | PE antimouse CD45 | BD Biosciences | Clone 30-F11 | (1:200) |
| antibody | PE antimouse CD31 | BD Biosciences | Clone MEC 13.3 | (1:200) |
| antibody | APC antimouse EpCAM | BD Biosciences | Clone EBA-1 | (1:200) |
| commercial assay or kit | Mouse Cytokine Array | R&D systems | ProteomeProfiler mouse Cytokine Array Panel A | |
| chemical compound, drug | LysoTracker | Molecular Probes | LysoTracker Green DND-26, Cat. Num. L7526 | |
| software, algorithm | MicroView | GE Healthcare | MicroView | |
| other | high-resolution CT system | GE Healthcare | CT Locus | |

## Mice and animal procedures

*Tert* heterozygous mice generated as previously described (*Liu et al., 2000*) were backcrossed to >98% C57/BL6 background. *Tert*$^{+/-}$ mice were intercrossed to generate first generation (G1) homozygous *Tert*$^{-/-}$ knock-out mice. G2 *Tert*$^{-/-}$ mice were generated by successive breeding of G1*Tert*$^{-/-}$. 8 to 10 weeks old male G2 *Tert*$^{-/-}$ mice were intratracheally inoculated with 0.5 mg/kg body weight bleomycin as previously described (*Povedano et al., 2015*). Mice within experimental groups were allocated randomly. Blind analysis of the samples was performed throughout this work.

All mice were produced and housed at the *specific pathogen-free* barrier area of the CNIO, Madrid. All animal procedures were approved by the CNIO-ISCIII Ethics Committee for Research and Animal Welfare (CEIyBA) (PROEX 42/13) and conducted in accordance to the recommendations of the Federation of European Laboratory Animal Science Associations (FELASA).

## Viral particle production

Viral vectors were generated as described (*Matsushita et al., 1998*) and purified as previously described (*Ayuso et al., 2014*). Vectors were produced through triple transfection of HEK293T. Cells were grown in roller bottles (Corning, NY, USA) in DMEM medium supplemented with fetal bovine serum (10% v/v) to 80% confluence and then co-transfected with the following plasmids: plasmid_1 carrying the expression cassette for gene of interest flanked by the AAV2 viral ITRs; plasmid_2 carrying the AAV *rep2* and *cap9* genes; plasmid_3 carrying the adenovirus helper functions (plasmids were kindly provided by K.A. High, Children's Hospital of Philadelphia). The expression cassettes were under the control of the cytomegalovirus (CMV) promoter and contained a SV40 polyA signal for *EGFP* and the CMV promoter and the 3'-untranslated region of the *Tert* gene as polyA signal for *Tert*. AAV9 particles were purified following an optimized method using two caesium chloride gradients, dialysed against PBS, filtered and stored at 80°C until use. Mice were injected via tail vein IV with 100 μL of rAVV9 viral genome particle ($2.5*10^{13}$ vg/mL).

## Histopathology, immunohistochemistry and immunofluorescence analysis

Histopathological analysis of paraffin-embedded lungs was performed in lung sections stained with nuclear fast red and Masson´s trichrome using standard procedures. To quantify collagen deposition picosirius red staining was performed on deparaffinised slides for 1 hr (*Broytman et al., 2015*).

Immunohistochemistry staining were performed with the following primary antibodies: rat monoclonal to p53 (POE316A/E9; CNIO histopathology core unit), rat monoclonal to p21 (HUGO-291H/B5; CNIO histopathology core unit), mouse monoclonal to phospho–Histone H2AX (Ser139) (Merck Millipore, Germany), rat monoclonal to F4/80 (ABD serotec, UK), p19ARF (5-C3-1 Santa Cruz Biotecchnology, Dallas, Texas), rabbit polyclonal to Sftpc (AB3786, Merck Millipore) and activated-caspase-3 (R&D systems, Minneapolis, Minesota).

For immunofluorescence, the antibodies used were goat polyclonal anti Sftpc (C-19; Santa Cruz Biotechnology), αSMA (CME 305; Biocare Medical, Concord, California), anti GFP (Roche, Switzerland) and rabbit monoclonal anti Ki67 (0003110QD; Master Diagnostica, Spain). Images were obtained using a confocal ultraspectral microscope ( TCS-SP5, Leica, Germany). Fluorescence intensities were analyzed with Definiens software. For each analysis, 3–5 lung sections and 10 visual fields/section were scored.

## In-vivo lung imaging by computed tomography (CT) and plethysmography

The acquisition was made on a high-resolution CT system (CT Locus, GE Healthcare) specially designed for small laboratory animals. Mice were anesthetized with a 4% rate of isoflurane (IsoVet Braun) during the induction and 2% during the maintenance period (scanning time). Micro-CT image acquisition consisted of 400 projections collected in one full rotation of the gantry in approximately 14 min in a single bed focused on the legs, with a 450 μA/80kV X-ray tube. 2-D and 3-D images were obtained and analysed using the software program MicroView (GE Healthcare). Pulmonary function was determined by plethymosgraphy using a pulmonary plethysmograph for sedated animals (Emka Technologies). The ratio between lung resistance and dynamic compliance (LR/Cdyn) was used as a measurement of pulmonary fitness. All procedures were carried out according to the European Normative of Welfare and Good Practice (2010/63/UE).

## Quantification of collagen peptides containing hydroxyproline

Proteins were extracted from lung samples in 8M urea/2M thiourea inTris pH 8.2 buffer using a Precelys disruptor and digested subsequently on a 30 KDa MWCO filter with LysC and Trypsin. Resulting peptides were resuspended in 1% TFA and desalted and concentrated using a homemade SCX Stage TIP (3M Empore). The samples were vacuum dried and dissolved in 100 μL of loading buffer (0.5% formic acid) and were analysed by LC-MS/MS in a Q-q-TOF Impact (Bruker Daltonics). The Impact was coupled online to a nanoLC Ultra system (Eksigent), equipped with a CaptiveSpray nano-electrospray ion source supplemented with a CaptiveSpray nanoBooster operated at 0.2 bar/minute with isopropanol as dopant. Samples (5 μL) were loaded onto a reversed-phase C18, 5 μm, 0.1 × 20 mm trapping column (NanoSeparations) and washed for 10 min at 2.5 μl/min with 0.1% FA. The

peptides were eluted at a flow rate of 250 nl/min onto an analytical column packed with ReproSil-Pur C18-AQ beads,2.4 µm, 75 µm x 50 cm (Dr. Maisch), heated to 45°C. Solvent A was 4% ACN in 0.1% FA and Solvent B acetonitrile in 0.1% FA. The gradient used was a 150 min curved gradient from 2% B to 33.2% B in 130 min. The MS acquisition time used for each sample was 150 min. The Q-q-TOF Impact was operated in a data dependent mode. The spray voltage was set to 1.35 kV (1868 nA) and the temperature of the source was set to 160°C. The MS survey scan was performed at a spectra rate of 2.5 Hz in the TOF analyzer scanning a window between 150 and 2000 m/z. The minimum MS signal for triggering MS/MS was set to a normalized threshold of 500 counts. The 20 most abundant isotope patterns with charge ≥2 and m/z > 350 from the survey scan were sequentially isolated and fragmented in the collision cell by collision induced dissociation (CID) using a collision energy of 23–56 eV as function of the m/z value. The m/z values triggering MS/MS with a repeat count of 1 were put on an exclusion list for 60 s using the rethinking option. For protein identification and quantification raw data were analyzed by MaxQuant interrogating a database containing mouse Uniprot Canonnical/TrEmbl sequences plus the most common contaminats (43936 entries), with Metionine and Proline Oxidation (HydroxyProline) allowed as variable modifications. A normalization factor was calculated to correct for variations in total protein content in each sample. Hydroxyproline-containing collagen peptides were quantified and their intensity values were normalized to the total peptide intensity for each sample.

## Cytokines array

Cytokine levels in lungs were analyzed by Mouse Cytokine Array (ProteomeProfiler mouse Cytokine Array Panel A from R&D Systems) following the manufacturer's instructions. The pixel density was determined by Image J Software.

## Telomere analysis

Q-FISH determination on paraffin-embedded tissue sections was performed as described previously (*González-Suárez et al., 2000*). After deparaffinization, tissues were post-fixed in 4% Formaldehyde 5 min, washed 3 × 5 min in PBS and incubated at 37°C 15 min in pepsin solution (0.1% Porcine Pepsin, Sigma; 0.01M HCl, Merck). After another round of washes and fixation as above-mentioned, slides were dehydrated in a 70%–90–100% ethanol series (5 min each). After 10 min of air-drying, 30 µl of telomere probe mix (10 mM TrisCl pH7, 25 mM MgCl2, 9 mM Citric Acid, 82 mM Na2HPO4, 70% Deionised Formamide –Sigma-, 0.25% Blocking Reagent –Roche- and 0.5 µg/ml Telomeric PNA probe -Panagene) were added to each slide. A cover slip was added and slides incubated for 3 min at 85°C, and for further 2 hr at RT in a wet chamber in the dark. Slides were washed 2 × 15 min in 10 mM TrisCl pH7, 0.1% BSA in 70% formamide under vigorous shaking, then 3 × 5 min in TBS 0.08% Tween20 and then incubated in a 4',6-diamidino-2-phenylindole (DAPI) bath (4 µg/ml DAPI (Sigma) in PBS) before mounting samples in Vectashield (VectorTM). Confocal image were acquired as stacks every 1 µm for a total of 3 µm using a Leica SP5-MP confocal microscope and maximum projections were done with the LAS-AF software. Telomere signal intensity was quantified using Definiens software.

## Gene expression analysis

RNA was extracted from post-caval lobe frozen lungs with RNeasy kit following manufacturer instruction (Qiagen, cat. N° 73504) and RNA integrity analyzed in an Agilent Bioanalyzer. cDNA was synthesised and analyzed on Agilent´s Mouse Genome DNA microarray, following the manufacturer´s instructions.

## Microarray analysis

Microarray background subtraction was carried out using normexp method. To normalize the dataset, we performed loess within arrays normalization and quantiles between arrays normalization. Differentially expressed genes were obtained by applying linear models with R limma package (Smyth GK) (Bioconductor project, http://www.bioconductor.org). To account for multiple hypotheses testing, the estimated significance level (p value) was adjusted using Benjamini and Hochberg False Discovery Rate (FDR) correction. Those genes with FDR < 0.05 were selected as differentially expressed between the AAV9-treated and non-treated groups. This standard FDR threshold assumes a 5% of

false positives in most impactful genes obtained in the differential expression analysis. The raw data have been deposited in GEO database (accession number GSE93869).

## Gene set enrichment analysis

Gene set enrichment analysis (GSEA) was applied using annotations from Biocarta, KEGG, NCI pathways and Reactome. Genes were ranked based on limma moderated t statistic. After Kolmogorov-Smirnoff testing, those gene sets showing FDR < 0.05, a well-established cut-off for the identification of biologically relevant gene sets (*Subramanian et al., 2005*), were considered enriched between classes under comparison.

## Flow cytometry

Cells were isolated from mouse lungs of both groups AAV9-*Tert* and AAV9-empty vector. Lungs were extracted and introduced in HBBS buffer with antibiotic and 1% BSA. Separate the lobules of the lung on a dish mince them with a scalpel. Transfer them to a GentleMacs tube with HBBS, antibiotics,, 1% BSA, DNAse I (60 units/mL) (Sigma, DN25) and collagenase type I (70 units/mL) (GIBCO, Cat. Number 17100). Then we run the GentleMac program 'lung 1', after the program incubate the sample at 37°C for 30 min and at the end we run the GentleMac program 'lung 2'. Cell suspension was filtered through a 40 µm stainer and then centrifuge 1200 rpm 5 min. Cells were resuspended in 2 mL ACK Lysis Buffer to lyse red blood cells. Incubate 4 min. at room temperature. We added DMEM without serum to wash, centrifugate and discard supernatant. At the end, resuspend cells in PBS with EDTA (1 mM), Hepes (25 mM) and 3% FBS to start staining with LysoTracker as described in commercial protocol (Molecular Probes, LysoTracker Green DND-26, Cat. Num. L7526) and the following antibodies from Pharmingen (BD Biosciences, San Jose CA): PE antimouse CD45, PE antimouse CD31, APC antimouse EpCAM. DAPI (Sigma, St Louis MO) was used to identify dead cells. Data was collected and the defined populations (CD45-CD31-EpCAM + LysoTracker + and Lyso-Tracker-) were sorted using an InFlux cell sorted (BD, San Jose CA), we excluded cell aggregates by using pulse processing in the scatter signals and dead cells in the basis of DAPI staining. All data was analyzed using FlowJo software v9.8.5 (Treestar, Ahsland OR).

# Acknowledgements

We are indebted to D Megias for microscopy analysis, to J Muñoz and F García for hydroxiproline analysis as well as to CNIO Histopathological Unit. The research was funded by project SAF2013-45111-R of Societal Changes Programme of the Spanish Ministry of Economics and Competitiveness (MINECO) co-financed through the European Fund of Regional Development (FEDER), *Fundación Botín* and Banco Santander (Santander Universities Global Division) and Roche Extending the Innovation Network Program (EIN) Academia Partnering Programme.

# Additional information

## Competing interests

Maria Bobadilla: is an employee for F. Hoffmann-La Roche Ltd, and the author declares no other competing financial interests. The other authors declare that no competing interests exist.

## Funding

| Funder | Grant reference number | Author |
| --- | --- | --- |
| Ministerio de Economía y Competitividad | SAF2013-45111-R | Paula Martinez |

The funders had no role in study design, data collection and interpretation, or the decision to submit the work for publication.

## Author contributions

Juan Manuel Povedano, Conceptualization, Validation, Investigation, Methodology, Writing—original draft; Paula Martinez, Conceptualization, Formal analysis, Supervision, Funding acquisition, Validation, Investigation, Methodology, Writing—original draft, Writing—review and editing; Rosa Serrano, Investigation, Methodology; Águeda Tejera, Validation, Investigation, Methodology; Gonzalo Gómez-López, Software, Formal analysis, Validation, Investigation, Methodology; Maria Bobadilla, Resources, Supervision, Investigation; Juana Maria Flores, Formal analysis, Investigation, Methodology; Fátima Bosch, Conceptualization, Supervision, Investigation, Methodology; Maria A Blasco, Conceptualization, Supervision, Funding acquisition, Investigation, Methodology, Project administration, Writing—review and editing

## Author ORCIDs

Juan Manuel Povedano (iD) http://orcid.org/0000-0002-8384-6540

Maria A Blasco (iD) http://orcid.org/0000-0002-4211-233X

## Ethics

Animal experimentation: Animal procedures were approved by the CNIO-ISCIII Ethics Committee for Research and Animal Welfare (CEIyBA) and conducted in accordance to the recommendations of the Federation of European Laboratory Animal Science Associations (FELASA).

## Decision letter and Author response

Decision letter https://doi.org/10.7554/eLife.31299.018

Author response https://doi.org/10.7554/eLife.31299.019

# Additional files

## Supplementary files

• Supplementary file 1. Differentially expressed genes in AAV9-*Tert* compared to empty vector treated fibrotic lungs (FDR < 0.05). The FDR is calculated by Benjamini and Hochberg FDR correction.

DOI: https://doi.org/10.7554/eLife.31299.012

• Transparent reporting form

DOI: https://doi.org/10.7554/eLife.31299.013

## Major datasets

The following dataset was generated:

| Author(s) | Year | Dataset title | Dataset URL | Database, license, and accessibility information |
|---|---|---|---|---|
| Povedano JM, Martinez P, Gomez-Lopez G, Blasco MA | 2018 | Therapeutic effects of telomerase in mice with pulmonary fibrosis induced by damage to the lungs and short telomeres | https://www.ncbi.nlm.nih.gov/geo/query/acc.cgi?acc=GSE93869 | Publicly available at the NCBI Gene Expression Omnibus (accession no: GSE93869). |

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
