## [Decision Letter]

Thank you for submitting your article "Therapeutic effects of telomerase in mice with pulmonary fibrosis induced by damage to the lungs and short telomeres" for consideration by *eLife*. Your article has been reviewed by three peer reviewers, and the evaluation has been overseen by Kathleen Collins as a guest Reviewing Editor and Jessica Tyler as the Senior Editor. The following individuals involved in review of your submission have agreed to reveal their identity: Dirk Hockemeyer (Reviewer #1).

The reviewers have discussed the reviews with one another and the Reviewing Editor has drafted this decision to help you prepare a revised submission.

Summary:

Short telomeres are a significant risk factor for the development of pulmonary fibrosis. Critically short telomeres impose an irreversible loss of alveolar type II (ATII) cells. This manuscript utilizes a short telomere mouse model (G2 *Tert^-/-^*) in which pulmonary fibrosis is induced by low dose bleomycin treatment. It examines the effect of upregulation of *Tert* expression via adeno-associated virus 9 (AAV9) delivery on fibrosis and associated functional, tissue, cellular, molecular, and gene expression changes. The results demonstrate overall improvement across several assays in the AAV9-*Tert* treated mice compared to the AAV9-empty controls.

The interest of this work for the *eLife* audience lies in its clinical relevance as proof of principle for telomerase-activation therapy of PF. Therapies to reverse pulmonary fibrosis by increasing telomere length are needed. Concerns were raised as to whether the mouse model of PF is close enough to the human disease versus another example of telomerase activation saving short-telomere mice from tissue insult. The authors need to explicitly justify the mouse model, noting that unlike the human disease, there is an improvement in the Bleo/AAV9-empty mice over time (e.g., Figure 1, Figure 2, and 3E). Thus the comparison made in this work is between improvement +/- *Tert* instead of improvement vs. no improvement that would be the case in human disease therapy.

Major revisions:

1) The paper would benefit from being streamlined towards the clinical model and relevance. The claims made from studying the ATII cells and the transcriptional profiling should be toned down, as proper lineage tracing is not performed.

2) It is unclear whether the sex of the mouse was taken into account. This is important, particularly, given their prior findings (Povedano, et al., 2015).

3) It is quite remarkable that approximately 80% of eGFP+ cells in the lung after AAV9-eGFP injection were ATII cells. Less than 20% of the ATII cells, however, were eGFP+. If AAV9 transduces 30% of lung cells, but ATII cells comprise ~60% of alveolar epithelial cells and 15% of peripheral lung cells, these numbers don't make sense. It would be good to know what% of lung cells are transduced and what percent of lung cells are ATII to understand the phenomenon reported in the paper. Moreover, it is also striking that transduction of such a minor fraction of ATII cells has the impact it does. This should be discussed.

4) The data in Figure 1 are difficult to interpret given each point in time is normalized to the corresponding measurement for the AAV9-empty control. A better comparison would be with the AAV9-empty control at week 0 as well as to mice not exposed to bleo but treated with AAV9-empty. The time points should also be carried out to 8 weeks post AAV9 treatment so one can determine whether the reversal of fibrosis as measured by CT and histology correlates with improvement in function.

5) CT appears to be a poor measure of fibrosis given that in the AAV9-empty mice less than 50% of the lung volume appears affected at 7 weeks (Figure 1), yet 100% of mice have severe fibrosis based on histology (Figure 1). Moreover, the CT (Figure 1) data suggest that a large percentage of G2 *Tert^-/-^* mice are able to recover to some extent following bleo treatment. This needs to be addressed.

6) At what time point were the mice used for Figure 2? As above, it is important to know how the measurements compare to mice that were mock-bleo treated followed by AAV9-empty. While there may be statistically significant differences between AAV9-empty and AAV9-*Tert* treated mice, it is important to know how the absolute values at any given time point differ from lung not treated with bleo.

7) There needs to be more clarity with respect to what AAV9-*Tert* treatment does with respect to fibrosis – does it enhance removal, decrease additional deposition, or a combination of both? Figure 2 suggest more rapid removal at the earlier time point.

8) The cytokine panels in Figure 2 need to be interpreted beyond there being a statistically significant difference between the AAV9-empty and AAV9-*Tert* mice at the two time points. There needs to be some interpretation as to whether the specific differences or lack thereof (e.g., M-CSF, TIMP-1, TNF-a) make sense. E.g., if macrophages are major drivers in pulmonary fibrosis does no change in M-CSF fit the proposed effect? If collagen removal is at play, does it make sense there was no change in TIMP-1? Do the authors have an idea as to why the hallmark inflammatory cytokine TNF-α is higher or similar for AAV9-empty mice and AAV9-*Tert* mice at the 3 week time point?

9) It is not clear what time point was used for Figure 3. This is important because Figure 3 shows marked reduction in p53 and p21 at 8 weeks even in the AAV9-empty. How does this correlate with the telomere and proliferation measurements?

10) The authors should show evidence of reactivation of telomerase in the *Tert*-AAV treated *Tert^-/-^* mice. This is particularly important as any difference in telomere length between AAV-empty and AAV-TERT treated mice is hard to discern (very little extension of telomeres may be necessary for a physiological effect).

11) The manuscript needs to be very carefully edited for spelling, grammar, and scientific writing. There are numerous errors throughout.

[Editors' note: further revisions were requested prior to acceptance, as described below.]

Thank you for resubmitting your work entitled "Therapeutic effects of telomerase in mice with pulmonary fibrosis induced by damage to the lungs and short telomeres" for further consideration at *eLife*. Your article was evaluated by the original peer reviewers, overseen by guest Reviewing Editor Kathleen Collins and Jessica Tyler as the Senior Editor.

The manuscript has been improved but there are some remaining issues that need to be addressed before acceptance, as outlined below:

All of the reviewers appreciate the revisions to the manuscript in content and in presentation. The concern that remains, after group discussion, whether there is actual reversal of fibrosis. The revised manuscript explains very clearly the criteria for scoring disease reversal, but (per reviewer discussion) "authors have not shown that they have reversed or treated PF in these mice" because they have not shown the specific pathology present at the time the mice are treated with AAV. The specific concerns are that perhaps only inflammation is present at that time, fibrosis ensues in the AAV-Empty but not the AAV-TERT mice, and the primary effect of AAV-TERT treatment is reversal of the inflammatory response. Nonetheless, the work demonstrates a significant benefit of AAV-TERT treatment in this model. The key word is "reversed", which is used three times in the text, one at the end of the Introduction after explaining what the diagnosis is that is reversed (so OK), then two at the very start of Results (header and first sentence) (also in the short title). If you could change this to something like "prevents pulmonary fibrosis progression and restores lung health" then the concerns are obviated. Perhaps it is possible to explain in the Discussion how humans present with the disease, but since humans and mice are not the same, this might not integrate well.

---

## [Author Response]

[…] Concerns were raised as to whether the mouse model of PF is close enough to the human disease versus another example of telomerase activation saving short-telomere mice from tissue insult. The authors need to explicitly justify the mouse model, noting that unlike the human disease, there is an improvement in the Bleo/AAV9-empty mice over time (e.g., Figure 1, Figure 2, and 3E). Thus the comparison made in this work is between improvement +/- Tert instead of improvement vs. no improvement that would be the case in human disease therapy.

Regarding the justification of the mouse model used here and previously developed by us (*Tert knock out* mice with short telomeres treated with a low bleomycin dose as a trigger of pulmonary damage), we previously showed that it recapitulated many aspects of the human disease including presence of short telomeres and development of a full-blown progressive disease (Povedano et al., 2015). In addition, we previously studied telomere length in other existing mouse models of PF to assess whether they could be used for the *Tert*-based gene therapy for the treatment of pulmonary fibrosis owing to short telomeres. On one hand, we analyzed telomere length in wild-type mice with pulmonary fibrosis owing to a standard bleomycin treatment, which is a widely-used mouse model for this disease (Adamson and Bowden, 1974). However, telomere length was not significantly affected in this mouse model, unlike the human IPF patients (Figure 1—figure supplement 1). We also tested telomere length in an additional mouse model that develops severe pulmonary fibrosis: the Fra2 knock-in mouse model (Eferl et al., PNAS, 2008) (Author response image 1). Again, Fra1-deleted mice diagnosed with pulmonary fibrosis did not show any significant telomere length changes in lung cells, unlike human PF patients with short telomeres. These previous results constituted the motivation and rational of why we ruled out these mouse models for our AAV9-*Tert* gene therapy studies. Instead, we intentionally chose *Tert*-knock out mice that present short telomeres and develop pulmonary fibrosis upon a low-dose bleomycin insult (Povedano et al., 2015). Thus, while wild-type mice with normal telomere length do not develop PF upon this low-dose bleomycin insult, the telomerase deficient mice presenting short telomeres develop PF, indicating that short telomeres are one of the underlying causes of PF. This is the only mouse model to our knowledge that develop PF as a consequence of telomere length defects. We now discuss these issues in the revised manuscript (Discussion, second paragraph). Finally, we would like to kindly indicate that, as in any other gene therapy trial, our approach consists in the reintroduction of a functional missing gene. In the case of pulmonary fibrosis owing to telomerase deficiency, this is a real clinical need, as there are a significant percent of patients suffering idiopathic pulmonary fibrosis associated to either telomerase mutations or to short telomeres (*Tert* and *Terc* account for 8-15% of familial and 1-3% of sporadic cases). Furthermore, up to 10% of patients with *no telomerase mutations* show telomere length in the range of mutation carriers, which could also benefit of a *Tert* gene therapy approach.

As per the second commentary by the Editor regarding Figure 1, we would like to highlight that even though in the longitudinal CT analyses shown in Figure 1 there is an apparent decrease of the affected area also in the empty-treated mice, given the small size of mouse lungs, CT imaging is not fully accurate to diagnose PF since inflammation can also give rise to abnormal CT pattern. For this reason we also performed longitudinal analyses of pulmonary function as determined by spirometry in these same mice. The results show that pulmonary function is not recovered in the AAV9-empty treated mice compared to the AAV9-*Tert* treated mice (new Figure 1). In addition, histopathological diagnosis of pulmonary fibrosis, clearly shows that while all of the AAV9-empty treated mice show severe fibrosis at the end-point of the experiment, 50% of the AAV9-*Tert* treated mice show complete remission of fibrosis and the other half show mild fibrosis (Figure 1), thus highlighting the beneficial effects of *Tert* treatment in this model of disease. Nevertheless,as suggested by the reviewer, we now state in the revised manuscript that AAV9-*Tert* therapy leads to more rapid improvement of PF as compared to AAV9-empty treated mice (Introduction: last paragraph; Discussion: third, sixth and last paragraphs).

**Author response image 1. respfig1:** Mean telomere length in wild type and fibrotic Fra2 Knock-in mice.

Major revisions:1) The paper would benefit from being streamlined towards the clinical model and relevance. The claims made from studying the ATII cells and the transcriptional profiling should be toned down, as proper lineage tracing is not performed.

We agree with the reviewer that the most important finding of our work is to provide a proof of principle that telomerase gene therapy constitutes a potential novel clinical treatment of pulmonary fibrosis associated with short telomeres. This conclusion is explicitly expressed in the text (Abstract; Discussion, last paragraph). We however think that the results obtained from the study of ATII are relevant as we previously demonstrated that ATII cells are a key cell type in the origin of pulmonary fibrosis owing to dysfunctional telomeres (Povedano et al., 2015). Nevertheless, we have now toned this down in the revised manuscript text by indicating that future lineage tracing experiments would be of interest (Discussion, sixth paragraph).

2) It is unclear whether the sex of the mouse was taken into account. This is important, particularly, given their prior findings (Povedano, et al., 2015).

All mice in this study were males. We have now stated this in the revised manuscript text (Results: page 7, 1st paragraph; page 9, 1^st^ paragraph; Methods: page 23, 1^st^ paragraph).

3) It is quite remarkable that approximately 80% of eGFP+ cells in the lung after AAV9-eGFP injection were ATII cells. Less than 20% of the ATII cells, however, were eGFP+. If AAV9 transduces 30% of lung cells, but ATII cells comprise ~60% of alveolar epithelial cells and 15% of peripheral lung cells, these numbers don't make sense. It would be good to know what% of lung cells are transduced and what percent of lung cells are ATII to understand the phenomenon reported in the paper. Moreover, it is also striking that transduction of such a minor fraction of ATII cells has the impact it does. This should be discussed.

We apologize for the confusion. We have now re-written this in the revised manuscript text (subsection “*Tert* targeting of alveolar type II cells reverses pulmonary fibrosis induced by short telomeres”, second paragraph). In the revised manuscript, we now include the data obtained in the current study on the percentage of transduced GFP+ cells upon AAV9-GFP delivery both with respect to total lung cells and with respect to total Sftpc+ ATII cells (see revised Figure 1). In particular, we find that 3% of total lung cells and 17.6% of ATII cells are transduced with AAV9-GFP (see revised Figure 1). In addition, by determining abundance of Sftpc-positive cells with regards to total lung cells we found that 13.4% of lung cells were ATII cells which is in agreement with the previously described 12-15% abundance of these cells in lungs (Dobbs, 1990; Van der Velden et al., 2013). Thus, our findings indicate that the large majority (79.5%) of lung cells transduced by AAV9 vectors correspond to ATII cells (cells positive for both Sftpc and for GFP with regards to total number of cells positive for GFP). The fact that AAV9 transduces ATII cells is of great importance as we have previously demonstrated that induction of severe telomere dysfunction specifically in these cells is sufficient to induce full-blown and lethal PF in mice (Povedano et al., 2015). As mentioned above, we have clarified this point in the revised manuscript. In particular, with approximately 17% of transduced ATII cells we observed a 2.5-fold increase in proliferation of global ATII cells (Figure 3) in AAV9-*Tert* treated mice compared to mice treated with the AAV9-empty vector and this was sufficient to clearly have beneficial effects on PF. We now discuss in the revised manuscript (Discussion, fourth paragraph).

4) The data in Figure 1 are difficult to interpret given each point in time is normalized to the corresponding measurement for the AAV9-empty control. A better comparison would be with the AAV9-empty control at week 0 as well as to mice not exposed to bleo but treated with AAV9-empty. The time points should also be carried out to 8 weeks post AAV9 treatment so one can determine whether the reversal of fibrosis as measured by CT and histology correlates with improvement in function.

As suggested by the reviewer, in the revised manuscript we now show absolute values (LR/Cdyn) for every single time point and include mice non-treated with bleomycin (new Figure 1). The results clearly show that AAV9-*Tert* treated mice present a similar pulmonary function as compared to healthy mice (non-treated with bleomycin) from two weeks onwards. In contrast, mice treated with AAV9-empty show significant higher LR/Cdyn values as compared to healthy and to AAV9-*Tert,* indicating a worsened pulmonary function (subsection “*Tert* targeting of alveolar type II cells reverses pulmonary fibrosis induced by short telomeres”, fourth paragraph). We thank the reviewer for this suggestion, which has clearly improved the figure and reinforced the message of the manuscript.

5) CT appears to be a poor measure of fibrosis given that in the AAV9-empty mice less than 50% of the lung volume appears affected at 7 weeks (Figure 1), yet 100% of mice have severe fibrosis based on histology (Figure 1). Moreover, the CT (Figure 1) data suggest that a large percentage of G2 Tert^-/-^ mice are able to recover to some extent following bleo treatment. This needs to be addressed.

We agree with the reviewer in that CT measurement of PF in mice is not the most accurate method to detect fibrosis due to the small size of mouse lungs and that inflammation can also give rise to an abnormal CT pattern. However, this technique is the only one that allows to perform an in vivo follow-up of the lungs. Nevertheless, the CT data show significant difference between the AAV9-*Tert* treated and AAV9-emptytreated mice at one week after IV injection. While in the AAV9-emptytreated mice the fibrotic volume increases, in the AAV9-*Tert* treated ones the fibrotic volume decreases. In addition, after the second week of treatment, we observed a regression of the affected CT lung volume in both groups, although at all time points analyzed the AAV9-*Tert* treated mice showed significantly smaller volume of the CT lesions as compared to mice treated with the empty vector. Nevertheless, we have now revised Figure 1 by comparing the affected areas at different times post-treatment to the initial area affected within each treatment group (see new Figure 1), which shows more clearly the therapeutic effect of *Tert* treatment compared to the treatment with the empty vector. As an additional way to longitudinally follow the effects of AAV9-*Tert* treatment on PF development, we also show spirometry data, again clearly showing an improvement in the AAV9-*Tert* treated group compared to mice treated with the empty vector (new Figure 1). We have also rephrased this point in the revised manuscript (subsection “*Tert* targeting of alveolar type II cells reverses pulmonary fibrosis induced by short telomeres”, fourth paragraph).

Importantly, in this work we assay fibrosis by several other different means than CT imaging; i.e. by quantification of hydroxiprolin containing collagen peptides (Figure 1), by Western blot of collagen at two different time-points after treatment (Figure 2), by histological methods (Masson’s trichrome and Sirius red stainings, Figure 1; Figure 2) and by immunofluorescence techniques for α-SMA quantification (Figure 2). We also assessed inflammation and expression of a wide panel of cytokines, as well as molecular markers of DNA damage, apoptosis and senescence at two different time points after treatment with the gene therapy vectors.

6) At what time point were the mice used for Figure 2? As above, it is important to know how the measurements compare to mice that were mock-bleo treated followed by AAV9-empty. While there may be statistically significant differences between AAV9-empty and AAV9-Tert treated mice, it is important to know how the absolute values at any given time point differ from lung not treated with bleo.

The time points at which the mice were analyzed in Figure 2 and throughout the manuscript have been clearly stated in the text and in figure legends.

The aim of this study was to address the therapeutic potential of *Tert* gene therapy as a novel treatment for PF. For that reason, we used as controls mice that received the same low-dose bleomycin treatment than AAV9-*Tert* treated mice but instead were treated with the AAV9-empty vector as placebo. Although we agree with the reviewer that mice without bleomycin AAV9-empty could also constitute an interesting control, we think that for our purposes Bleo/AAV9-empty is the most informative control. Including at this stage a non-bleomycin treated group would greatly delay publication of our results without significantly changing the main findings of our manuscript on the effectivity of *Tert* gene therapy in our mouse model of PF.

7) There needs to be more clarity with respect to what AAV9-Tert treatment does with respect to fibrosis – does it enhance removal, decrease additional deposition, or a combination of both? Figure 2 suggest more rapid removal at the earlier time point.

We agree with the reviewer that the data we have indicate that telomerase expression leads to a more rapid removal of fiber deposition as compared to AAV9-treated mice. This is an interesting point and we have discussed it in the revised manuscript text (subsection “*Tert* targeting of alveolar type II cells reverses pulmonary fibrosis induced by short telomeres”, seventh paragraph).

8) The cytokine panels in Figure 2 need to be interpreted beyond there being a statistically significant difference between the AAV9-empty and AAV9-Tert mice at the two time points. There needs to be some interpretation as to whether the specific differences or lack thereof (e.g., M-CSF, TIMP-1, TNF-a) make sense. E.g., if macrophages are major drivers in pulmonary fibrosis does no change in M-CSF fit the proposed effect? If collagen removal is at play, does it make sense there was no change in TIMP-1? Do the authors have an idea as to why the hallmark inflammatory cytokine TNF-α is higher or similar for AAV9-empty mice and AAV9-Tert mice at the 3 week time point?

The aim to analyze a large panel of cytokines at two different time points, 3 and 8 weeks post AAV9 inoculation, was to get an overall insight of the inflammatory status of the AAV9-*Tert* compared to the empty treated lungs during the viral treatment. The results clearly show decreased levels of almost all the cytokines analyzed at both time-points in AAV9-*Tert* lungs, indicating less inflammation. Several lines of evidence implicate cytokines in fibrosis. Abnormal cytokine expression in fibrotic tissues indicate association between a cytokine and the fibrotic process. However, such changes do not necessarily implicate the cytokine as a contributor to the fibrotic process, as they can be a result rather than a cause of the disease. To implicate specific cytokines as a casual factor of fibrosis more extensive work is needed. We think we have not sufficient data to draw any conclusion about the potential roles of specific cytokines in the observed PF improvement in AAV9-*Tert* treated mice. Therefore, we rather show our cytokine analysis as a descriptive observation indicating less inflammation in the AAV9-*Tert* as compared to control treated lungs.

9) It is not clear what time point was used for Figure 3. This is important because Figure 3 shows marked reduction in p53 and p21 at 8 weeks even in the AAV9-empty. How does this correlate with the telomere and proliferation measurements?

As suggested by the reviewers, the time points at which the mice were analyzed in Figure 3 are now indicated in the revised text and revised figure legend. The higher number of proliferating ATII cells are in agreement with the significant lower percentage of short telomeres in ATII cells in AAV9-*Tert* treated lungs as well as with the significant decrease in the number of p21 and p53 positive cells in AAV9-*Ter*t treated lungs (see Figure 3). This has been now discussed in the revised manuscript (subsection “AAV9-*Tert* treatment results in increased proliferation of ATII cells”, last paragraph).

10) The authors should show evidence of reactivation of telomerase in the Tert-AAV treated Tert^-/-^ mice. This is particularly important as any difference in telomere length between AAV-empty and AAV-TERT treated mice is hard to discern (very little extension of telomeres may be necessary for a physiological effect).

In the manuscript, we show a clear transcriptional expression of TERT in the telomerase deficient mice treated with AAV9-*Tert* while this expression is undetectable in AAV9-empty treated samples (Figure 1 and Figure 5). We also show a 10% increase in telomere length and a significant reduction in the percentage of short telomeres in ATII cells of AAV9-*Tert* as compared to AAV9-empty treated mice (Figure 3). Regarding reactivation of telomerase in the adult lungs upon AAV9-*Tert* treatment, we have demonstrated it in a previous paper (Bernardes de Jesus et al., 2012). We now cite this in the revised manuscript (subsection “*Tert* targeting of alveolar type II cells reverses pulmonary fibrosis induced by short telomeres”, second paragraph).

11) The manuscript needs to be very carefully edited for spelling, grammar, and scientific writing. There are numerous errors throughout.

We have carefully edited the revised manuscript

[Editors' note: further revisions were requested prior to acceptance, as described below.]

All of the reviewers appreciate the revisions to the manuscript in content and in presentation. The concern that remains, after group discussion, whether there is actual reversal of fibrosis. The revised manuscript explains very clearly the criteria for scoring disease reversal, but (per reviewer discussion) "authors have not shown that they have reversed or treated PF in these mice" because they have not shown the specific pathology present at the time the mice are treated with AAV. The specific concerns are that perhaps only inflammation is present at that time, fibrosis ensues in the AAV-Empty but not the AAV-TERT mice, and the primary effect of AAV-TERT treatment is reversal of the inflammatory response. Nonetheless, the work demonstrates a significant benefit of AAV-TERT treatment in this model. The key word is "reversed", which is used three times in the text, one at the end of the Introduction after explaining what the diagnosis is that is reversed (so OK), then two at the very start of Results (header and first sentence) (also in the short title). If you could change this to something like "prevents pulmonary fibrosis progression and restores lung health" then the concerns are obviated. Perhaps it is possible to explain in the Discussion how humans present with the disease, but since humans and mice are not the same, this might not integrate well.

As suggested by the reviewers, we have changed the word “reversed” by “prevents the progression of pulmonary fibrosis and restores lung health” in the text. The short title has been replaced by “Prevention of pulmonary fibrosis progression by telomerase.